# CFTR modulator therapy drives microbiome restructuring through improved host physiology in cystic fibrosis: the IMMProveCF phase IV trial

Cystic fibrosis (CF) is a genetic disorder caused by mutations in the CFTR gene, leading to impaired CFTR function, mucus accumulation, chronic infections, and inflammation. The triple combination elexacaftor/tezacaftor/ivacaftor (ETI) has transformed CF treatment by restoring CFTR function. However, how ETI-induced physiological improvements affect long-standing dysbiosis and pathogen colonization across microbiome habitats remains poorly understood. In this prospective longitudinal study (DRKS00023862), we analyzed sputum, throat, and stool microbiomes of pwCF (n = 35) before and after ETI initiation, alongside healthy controls (n = 49). The primary endpoint was longitudinal change in diversity, species richness, and microbial composition in the respiratory and intestinal microbiome, profiled by 16S rRNA gene sequencing. Secondary endpoints included changes in lung function, systemic and gastrointestinal inflammation. We show how improved CFTR function and direct antibacterial effects of ETI create a niche disadvantage for *Staphylococcus* in the sputum microbiome. Respiratory microbiome shifts were immediate, while gut changes emerged gradually. *Escherichia* abundance in stool, initially elevated in pwCF, decreased post-ETI and correlated with lower fecal calprotectin. These findings demonstrate that ETI can partially reverse CF-associated dysbiosis through improved host physiology. They offer insights into host-microbiome dynamics under therapeutic modulation and emphasize the need for confounder-aware models in complex clinical populations.

Cystic fibrosis (CF) is a life-threatening condition that primarily affects the respiratory, digestive, and reproductive systems[1]. CF results from mutations in the CF transmembrane conductance regulator (CFTR) gene. Its encoded protein functions as a chloride channel, crucial for regulating ion and water balance in epithelial tissues. Mutations disrupt this function, leading to the production of thick, sticky mucus that can obstruct airways, ducts, and other passageways in the body. This creates a favorable environment for bacterial lung infections, chronic inflammation, and ultimately, progressive lung damage[2]. Prior to the introduction of CFTR modulators in CF health care, people with CF (pwCF) were highly medicated with antibiotics to combat recurrent or chronic infections, by this ultimately aggravating CF-linked dysbiosis and selection for pathogens such as *Staphylococcus aureus* and *Pseudomonas aeruginosa*.

Since its approval in 2019, the triple CFTR modulator therapy elexacaftor/tezacaftor/ivacaftor (ETI) has transformed CF care by

e-mail: krpoplawska@icloud.com; sofia.forslund@mdc-berlin.de

improving lung function and clinical outcomes in pwCF carrying specific mutations, including the common F508del allele[3,4]. ETI improves CFTR protein function, and while its benefits for lung health are well-documented, its effects on microbial dysbiosis, prevalent before treatment, remain less understood[2,5]. Several recent studies have reported similar microbiome transitions following ETI therapy (Supplementary Table S1), although most were single-center with short follow-up and focused on either the respiratory or gut microbiome in isolation.

Sputum microbial dysbiosis has been linked to reduced lung function, a marker for CF severity[6]. Although studies have shown a general increase in sputum microbial diversity during the initial 3–12 months of ETI treatment[7,8], their persistence, and their relationship to host physiology or medication remain unclear. Beyond the lungs, dysbiosis in the gastrointestinal tract is well documented in pwCF, marked by reduced microbial diversity and an increase in pathobionts and pro-inflammatory bacteria[9]. Interestingly, ETI's effects on the gut microbiome remain contradicting as diversity increased in children after six months of treatment[10] but declined in adults[11]. The mechanisms underlying these contrasting patterns remain largely unexplored.

The gut microbiome is highly responsive to drugs[12], highlighting the importance of accounting for confounding effects to uncover true host-microbiome associations[13]. This is particularly relevant as CFTR modulators have been suggested to exhibit direct antibacterial effects, with ivacaftor, inhibiting the growth of gram-positive bacteria[14] and potentiating antibacterial effects on *S. aureus*[15]. Studying the microbiome across multiple habitats in pwCF, who experience multimorbidity and extensive medication use, therefore presents both challenges and opportunities for uncovering host-microbiome interactions in vivo.

We hypothesized that ETI modulates the microbiome both indirectly, through improvements in host physiology, and directly via its intrinsic antibacterial properties. To test this, we longitudinally sampled sputum, throat, and stool from 35 pwCF at 3-month intervals for up to 24 months following ETI initiation. Our primary objective was to characterize microbiome dynamics in response to ETI across these distinct body habitats. As a secondary objective, we assessed concurrent changes in clinical parameters, including inflammation markers and antibiotic use.

Here, we show that ETI induces marked changes in the respiratory and gastrointestinal microbiomes. While the respiratory microbiome responds rapidly to ETI, marked by a decline in *S. aureus* mediated by improved CFTR-function, the gut microbiome shows a delayed response, primarily driven by reduced systemic antibiotic use and decreased gastrointestinal inflammation. Our in vitro assays reveal that potentially pathogenic species such as *Escherichia coli*, *Enterococcus faecium*, and *P. aeruginosa* are less susceptible to the direct antimicrobial effects of ETI than other microbiome members, which show increased sensitivity. Beyond its implications for CF treatment, our study provides a framework for understanding how physiological changes influence microbiome dynamics and explores to which extent established microbial dysbiosis can be reversed, offering insights applicable to other disease conditions.

## Results

The study enrolled 35 pwCF, aged 6-55 years, from a single CF outpatient clinic in Germany. Cohort characteristics are detailed in Table 1. Children aged 6-11, who were enrolled mid-study, after ETI approval was extended to this age group, contributed samples limited to the first 15 months of treatment. Additionally, 49 healthy, age- and sex-matched individuals were included as controls.

Of the 35 pwCF, 18 provided baseline sputum samples, with 11 contributing at least one follow-up sample, totaling 57 sputum samples (Fig. 1a, and Supp. Fig. 1a). A reduction in sputum production following ETI treatment is commonly observed as lung health improves[16,17],

though this may be linked with younger age and fewer respiratory co-morbidities. In our cohort, pwCF who provided follow-up sputum samples were generally older and had lower lung function compared to those who did not provide any sputum samples (Table 2). Additionally, 252 throat and 251 stool samples were collected from both pwCF and controls (Fig. 1a, Supp. Figure 1a-d). In addition to these biosamples, comprehensive clinical data were collected to evaluate pwCF's physiological responses to the new treatment and any adjustments made to chronic medication regimens.

### Distinct clinical improvements in response to ETI treatment

To account for variability within our heterogeneous cohort, we employed linear mixed effect models (LME) adjusted for sex and age to analyze the collected data. This approach confirmed that the observed changes were not artifacts of imbalanced sampling across participant subgroups (Table 2). While age and sex significantly impacted several clinical parameters (as detailed below), their effect on microbiome composition was limited. Specifically, age was only associated with reduced microbial richness in sputum (N observed ASVs), with no significant impact observed on throat or stool alpha diversity, nor overall microbial community structure (Bray-Curtis (BC) dissimilarity; PERMANOVA) across habitats. In contrast, ETI and other clinical factors drove significant improvements in both clinical parameters and microbiome composition compared to baseline (Fig. 1 and Supp. Fig. 2).

Specifically, we observed a notable increase in lung function (ppFEV1) with ETI treatment, with median improvements ranging from 6% to 15% (LME, FDR ≤ 0.01 across all time points compared to baseline). Additionally, sweat chloride levels showed a substantial reduction, with median decreases of 41 to 44 mmol/L (LME, FDR ≤ 0.001 at 3, 12, 18, and 24 months of ETI treatment compared to baseline). In several cases reaching levels < 30 mmol/l, which are considered healthy.

We also found participants had fewer days of antibiotic treatment, with a median reduction of up to 35 days compared to the participant's antibiotic exposures in the year prior to ETI initiation (LME, FDR ≤ 0.001 at 9 months and thereafter). Systemic inflammation markers decreased significantly, with IgG levels showing median reductions of −1.0 to −2.4 g/L compared to baseline (LME, FDR ≤ 0.001 at 3-21 months of ETI treatment, and FDR ≤ 0.01 at 24 months). IL-6 levels also exhibited consistent decreases with a median reduction between 7-12.5 mmol/l across all months of ETI treatment (LME, FDR ≤ 0.1 compared to baseline). However, no significant changes were observed in leukocyte counts or IL-8 levels.

Fecal calprotectin levels declined steadily from a median of 147 µg/g at baseline to a median of 22-79 µg/g across subsequent visits, indicating reduced intestinal inflammation (LME, FDR ≤ 0.01 for all time points vs. baseline). Reduction in HbA1c (ranging in median −0.05 to −0.55%, LME FDR ≤ 0.05 at 6,9,12 and 18 months after ETI treatment start) suggests improved pancreatic endocrine function. As a treatment side effect, Alanine Aminotransferase (ALAT) increased by median 4-7 µl/l (LME FDR ≤ 0.05 at 3-12 months compared to baseline), but Aspartate Aminotransferase (ASAT) levels remained stable. This ALAT increase was not significantly different compared to baseline in the second year of treatment, aligning with mild increases as reported previously[18].

Age and sex significantly (FDR ≤ 0.05) influenced several studied parameters. Female and older participants had more antibiotic treatment days. Older participants exhibited lower lung function (ppFEV1, ppFVC) and higher IgG levels, while females showed lower IL-6 and ASAT levels but higher ALAT levels. Detailed statistical results are provided in Supplementary Data 1 and 2.

ETI treatment led to consistent clinical improvements, including better lung function, reduced sweat chloride, lower inflammation, fewer antibiotic days, and minor, temporary increases in liver enzymes.

**Table 1 | Cohort characteristics**

| | Baseline N = 35 | 3 months N = 30 | 6 months N = 31 | 9 months N = 31 | 12 months N = 26 | 15 months N = 23 | 18 months N = 22 | 21 months N = 22 | 24 months N = 19 | Controls N = 49 |
|---|---|---|---|---|---|---|---|---|---|---|
| **Sex** | | | | | | | | | | |
| Female n (%) | 18 (51%) | 15 (50%) | 16 (52%) | 15 (48%) | 12 (46%) | 13 (57%) | 12 (55%) | 12 (55%) | 9 (47%) | 29 (59%) |
| **Age in years** | | | | | | | | | | |
| Median (IQR) | 24 (13, 31) | 24 (12, 30) | 25 (13, 33) | 25 (14, 31) | 27 (14, 32) | 28 (23, 35) | 29 (23, 36) | 28 (24, 34) | 30 (24, 43) | 24 (18, 33) |
| min-max | 6–53 | 6–53 | 6–53 | 6–53 | 6–54 | 9–54 | 13–54 | 13–54 | 14–55 | 6-56 |
| **Age groups** | | | | | | | | | | |
| **>=20** n (%) | 23 (66%) | 20 (67%) | 20 (65%) | 21 (68%) | 18 (69%) | 18 (78%) | 17 (77%) | 17 (77%) | 15 (79%) | 32 (65%) |
| **12-19** n (%) | 4 (11%) | 4 (13%) | 5 (16%) | 5 (16%) | 5 (19%) | 4 (17%) | 5 (23%) | 5 (23%) | 4 (21%) | 13 (27%) |
| **<=11** n (%) | 8 (23%) | 6 (20%) | 6 (19%) | 5 (16%) | 3 (12%) | 1 (4.3%) | 0 (0%) | 0 (0%) | 0 (0%) | 4 (8.2%) |
| **Mutation** | | | | | | | | | | |
| **F508del homozygous** n (%) | 26 (74%) | 23 (77%) | 24 (77%) | 23 (74%) | 19 (73%) | 18 (78%) | 18 (82%) | 17 (77%) | 16 (84%) | |
| **Lung function** | | | | | | | | | | |
| **ppFEV1** Median (IQR) | 82 (68, 99) | 97 (85, 109) | 98 (82, 105) | 97 (78, 108) | 87 (75, 105) | 88 (75, 100) | 84 (73, 102) | 81 (68, 103) | 79 (69, 91) | |
| **ppFVC** Median (IQR) | 94 (84, 106) | 107 (98, 121) | 103 (97, 113) | 102 (98, 114) | 102 (95, 111) | 103 (95, 111) | 102 (93, 108) | 102 (93, 111) | 98 (90, 104) | |
| **Sweat chloride** | | | | | | | | | | |
| **[mmol/l]** Median (IQR) | 86 (79, 96) | 50 (42, 56) | | | 44 (32, 59) | | 50 (27, 56) | | 46 (35, 53) | |
| **Bacterial culture results**[1] | | | | | | | | | | |
| **Sputum** | | | | | | | | | | |
| *Staphylococcus aureus* pos. n (%) | 11 (61%) | 4 (40%) | 3 (60%) | 5 (71%) | 1 (20%) | 1 (17%) | 2 (40%) | 2 (40%) | 1 (17%) | |
| *Pseudomonas aeruginosa* pos. n (%) | 10 (56%) | 4 (40%) | 2 (40%) | 2 (29%) | 3 (60%) | 2 (33%) | 2 (40%) | 2 (40%) | 1 (17%) | |
| **No culture results** n (%) | 17 (49%) | 20 (69%) | 26 (84%) | 24 (77%) | 21 (81%) | 17 (74%) | 17 (77%) | 17 (77%) | 13 (68%) | |
| **Throat** | | | | | | | | | | |
| *Staphylococcus aureus* pos. n (%) | 17 (71%) | 20 (69%) | 19 (66%) | 17 (59%) | 11 (44%) | 9 (45%) | 11 (55%) | 11 (50%) | 6 (32%) | |
| *Pseudomonas aeruginosa* pos. n (%) | 5 (21%) | 5 (17%) | 3 (10%) | 4 (14%) | 2 (8.0%) | 3 (15%) | 3 (15%) | 4 (18%) | 5 (26%) | |
| **No culture results** n (%) | 11 (31%) | 1 (3%) | 2 (6%) | 2 (6%) | 1 (4%) | 3 (13%) | 2 (9%) | 0 (0%) | 0 (0%) | |

[1] We compared bacterial culture results to detection of *Pseudomonas* and *Staphylococcus* with 16S rRNA gene sequencing. Accuracy between methods ranged from 90% for *Pseudomonas* in throat samples to 54% for *Staphylococcus* in throat samples, results are summarized in Supplementary Data 15.

## Loss of *Staphylococcus* dominance drives consistent shifts in sputum microbial diversity and composition following ETI treatment

To evaluate the impacts of the ETI treatment on the microbiome, we first quantified the bacterial load in samples via 16S marker gene qPCR and detected no changes compared to baseline (Supp. Fig. 1e).

Eighteen CF participants provided baseline sputum samples, 11 of whom provided at least one sputum sample after the start of ETI treatment (3 participants with 1 follow-up sample, 9 participants with more than 2 follow-up samples; Supp. Figs. 1a, b, 3f). Among these participants with follow-up sputum samples, we observed a significant shift in bacterial composition as early as 3 months after the initiation of ETI treatment, as measured by BC dissimilarity (PERMANOVA stratified by donor: Baseline vs. 3-month samples: $R^2 = 7.4\%$, p = 0.02; Figs. 2a, b, and Supplementary Data 3). No significant shifts in microbial composition were detected when comparing samples collected 6 months or later (Fig. 2b: PERMANOVA 6-12 months vs 15–18 months: $R^2 = 2.6\%$, p = 0.54; 6–12 months vs 21–24 months: $R^2 = 2.8\%$, p = 0.58), suggesting that the most pronounced changes occur within the first months of treatment, after which a "new stable state" is established.

Within individual samples, we observed a significant increase in both sputum Shannon diversity and microbial richness, with this increase becoming significant in samples collected 6 months or later when compared to baseline (LME Baseline vs 6-12 months: Estimate=0.7, FDR = 0.005, estimate=26.8 and FDR = 0.001, respectively; Supp. Figs. 2a, b, full statistical results on alpha diversity measures in Supplementary Data 4).

We further investigated taxonomic-level changes and found in this cohort a complete loss of *Staphylococcus* dominance (Fig. 2c), accompanied by a significant reduction in *Staphylococcus* and *Staphylococcaceae* relative abundance (Supp. Fig. 3c, LME Baseline vs 15-18 months: Estimate = −0.7, FDR = 0.047, Estimate = −0.7, FDR = 0.026, respectively).

While 6 out of 11 baseline samples were *Staphylococcus*-dominated, none of the follow-up samples exhibited this pattern. Instead, microbial compositions shifted towards *Streptococcus*, *Rothia*, or *Pseudomonas* dominance. Notably, 2 participants exhibited *Pseudomonas*-dominated sputum microbiota post-ETI initation, neither of whom had *Staphylococcus* or *Pseudomonas* dominance at baseline. Additionally, 2 of the 6 participants who were initially *Staphylococcus*-

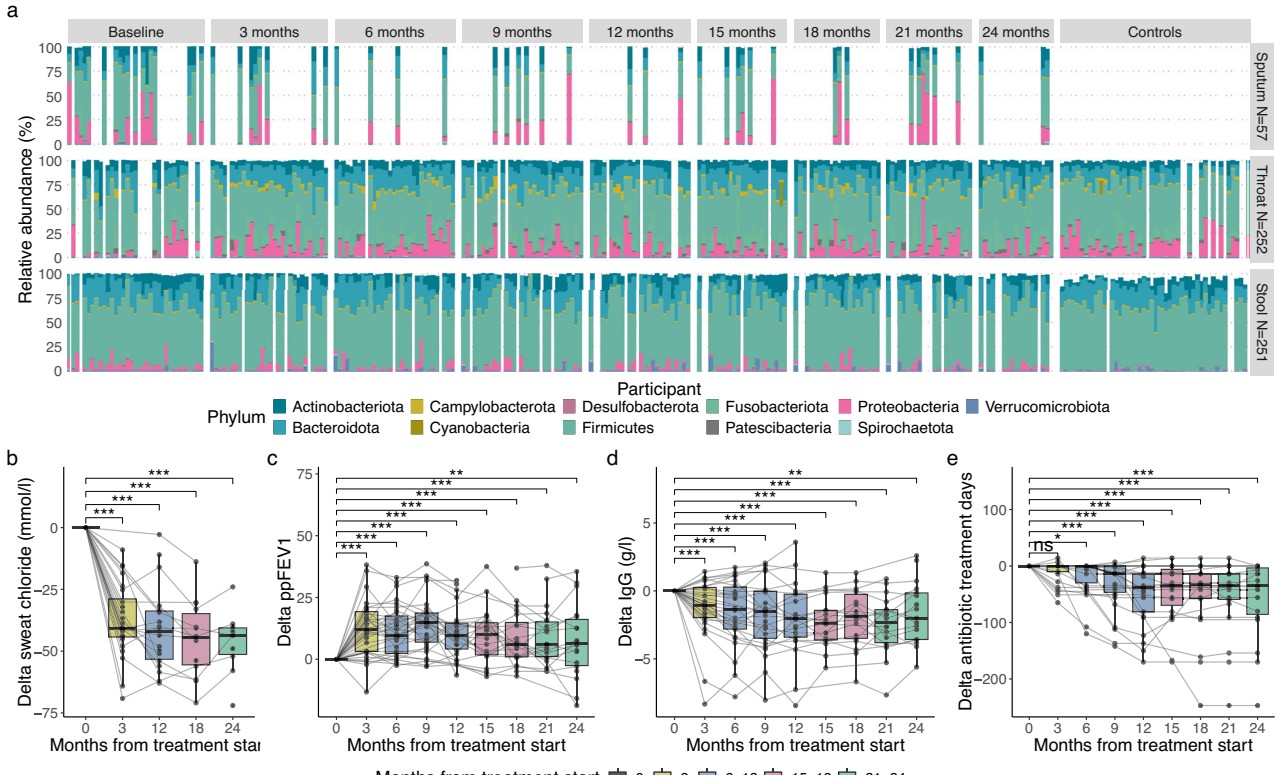

**Fig. 1 | Longitudinal microbiome composition and clinical changes in response to ETI treatment. a** Overall taxonomic composition at phylum level per sample type for all participants during 24 months. Only phyla with a relative abundance >2% across the dataset are displayed. Samples within the same column of the x-axis correspond to the same participant, the numbers and types of samples provided per time point differ between participants. Throat and stool samples from controls were collected at a single time point. **b** Reduction in sweat chloride from baseline at 3, 12, 18, 24 months from ETI treatment initiation. **c** Increase in lung function (ppFEV1 = predicted percent forced expiratoy volume in 1 second) from baseline per sample time point. **d** Decrease in serum IgG levels detected from baseline per sample time point. **e** Decrease in the number of days with systemic (oral or intravenous) antibiotic treatment from baseline per sample time point. Box plots show the median (line), IQR (box), 1.5× IQR range (whiskers). Points represent individual samples, lines connect intra-individual follow-ups to highlight trajectories, asterisks indicate significance in linear mixed effects models with id as a random factor, corrected for sex and age (FDR*** <= 0.001, FDR** <= 0.01, FDR* <= 0.05, FDR$^{ns}$ >=0.1). Colors indicate months from ETI treatment start: 3 months=yellow, 6–12 months=blue, 15–18 months=red, 21–24 months=green. The corresponding N of observations per sample time point and LME results are available in Supplementary Data 1.

dominated showed an increased relative abundance of *Pseudomonas* post-treatment, although without reaching dominance (Supp. Fig. 3f).

In our cohort, ETI treatment lead to a rapid and lasting shift in sputum microbiome composition, marked by a significant loss of *Staphylococcus* dominance within the first 3 months, followed by increased microbial diversity and a new stable community structure dominated by *Streptococcus*, *Rothia*, or *Pseudomonas*.

### Sputum microbiome dynamics reflect medication and physiological changes

Given the significant host physiological changes and alterations in routine medication following ETI treatment, particularly the reduction in antibiotic use, we conducted a confounder/covariate-aware analysis to determine which clinical parameters were most strongly associated with shifts in the sputum microbiome.

To assess how individual covariates contributed to the variation in microbial community composition, measured by BC dissimilarity, we applied a two-step approach. First, we tested each covariate independently using simple PERMANOVA, stratified by donor, to evaluate its marginal contribution to the variance. In addition to sampling time point and total ETI treatment duration (in days), sweat chloride, fecal calprotectin, inhaled DNAse treatment, and the liver enzyme GPT were found to significantly explain the variance in sputum microbiome composition (Fig. 2d).

In the second step, we performed a multiple PERMANOVA stratified by donor (N = 31 samples), combining the identified significant clinical covariates. Sweat chloride was collinear with ETI treatment duration (Variance Inflation Factor, VIF = 4.5), but DNAse treatment was independent of it. As a result, sweat chloride explained significant variance only when ETI treatment time was excluded from the multiple PERMANOVA. After excluding ETI treatment time, sweat chloride ($R^2$ = 11.2%, $p$ = 0.009) and DNAse treatment ($R^2$ = 4.9%, p = 0.03) remained as significant contributors to microbiome variance.

To examine how clinical factors and medications influence specific microbial taxa and alpha diversity measures, we used the metadeconfoundR package[19]. This approach applies nested linear model comparisons to account for confounding effects, ensuring that only associations statistically significant after deconfounding are reported (Supp. Fig. 3c). This analysis confirmed that *Staphylococcus* relative abundance decreased with ETI treatment duration (metadeconfoundR, Spearman's rho = −0.56, FDR = 0.009; Confounder: sweat chloride). However, this association was confounded by sweat chloride levels, which strongly correlated with the reduction in *Staphylococcus* (metadeconfoundR, Spearman's rho = 0.57, FDR = 0.015, Fig. 2e). Further validation through mediation analysis revealed that 99.8% of the decrease in *Staphylococcus* abundance was mediated by changes in sweat chloride levels (p = 0.01) (Fig. 2f), highly suggesting that the

**Table 2 | Cohort Characteristics at baseline between subgroups: stratified by sputum availability and follow-up status**

| Characteristics at Baseline | Sputum with FU N = 11 | Sputum no FU N = 7* | No Sputum N = 16 | Statistics |
|---|---|---|---|---|
| **Sex** | | | | |
| **Female** n (%) | 5 (45%) | 5 (71%) | 9 (56%) | FT: $p = 0.38$ |
| **Age in years** | | | | |
| Median (IQR) | 26(23,32) | 28 (21, 30) | 16 (9, 28) | KW: $\chi^2 = 4.5$, $p = 0.11$ |
| **Age groups** | | | | |
| **>=20** n (%) | 10 (91%) | 5 (71%) | 8 (50%) | FT: **$p = 0.03$**; Spu NFU - NoSpu: $p = 0.17$; Spu FU - NoSpu: **$p = 0.03$**; Spu FU - Spu NFU: $p = 0.35$ |
| **12-19** n (%) | 1 (9.1%) | 1 (14%) | 1 (6.3%) | |
| **<=11** n (%) | 0 | 1 (14%) | 7 (44%) | |
| **Mutation** | | | | |
| **F508del homozygous** n (%) | 9 (82%) | 5 (71%) | 12 (75%) | FT: $p = 0.7$ |
| **Lung function** | | | | |
| **ppFEV1** Median (IQR) | 69 (52, 76) | 79 (69, 83) | 94 (82, 99) | KW: $\chi^2 = 6.5$, **$p = 0.04$**; Spu NFU - NoSpu W: $p = 0.18$; Spu FU - NoSpu W: **$p = 0.02$**; Spu FU - Spu NFU W: $p = 0.23$ |
| **ppFVC** Median (IQR) | 84 (80, 96) | 94 (91, 100) | 100 (93, 115) | KW: $\chi^2 = 4.2$, $p = 0.12$ |
| **Sweat chloride** | | | | |
| **[mmol/l]** Median (IQR) | 85 (77, 95) | 97.2 (88.5, 99.4) | 82 (78, 93) | KW: $\chi^2 = 4.8$, $p = 0.09$ |
| **Bacterial culture results in sputum** | | | | |
| *Staphylococcus aureus* pos. n (%) | 7 (64%) | 4 (57%) | NA | FT: $p = 1$ |
| *Pseudomonas aeruginosa* pos. n (%) | 7 (64%) | 3 (43%) | NA | FT: $p = 0.6$ |

*FU* follow-up, *N* number of participants, *n(%)* number and percentage of participants, *IQR* interquartile range, *ppFEV1* percent predicted forced expiratory volume in 1 second, *ppFVC* percent predicted forced vital capacity. Statistical tests used: *FT* Fisher's exact test, *KW* Kruskal-Wallis test, *W* Wilcoxon rank-sum test for post-hoc pairwise comparisons where applicable. Significant *p*-values are depicted in bold. NA = not applicable (no sputum sample available for microbiological assessment).
* 1 patient (IMP8) provided a single sputum sample at Visit 9, but no baseline sample, and is not accounted for in this subgroup analysis.

observed reduction in *Staphylococcus* is driven by improved CFTR function in response to ETI treatment.

*Pseudomonas* relative abundance correlated positively with leukocyte count and negatively with lung function (metadeconfoundR: leukocytes: Spearman's rho = 0.6, FDR < 0.001; ppFEV1: Spearman's rho = −0.33, FDR = 0.034; Supp. Figs. 3c and 4), with no significant effect of ETI treatment duration observed. Subgroup analysis showed a significant reduction in leukocyte counts with ETI in *Pseudomonas*-negative participants, though not in the total cohort, highlighting the sputum microbiome's influence on clinical trajectories and ETI response (Supp. Fig. 4).

Additionally, we observed an increase in the relative abundance of several sputum microbiome members—such as *Neisseria*, *Eubacterium brachy* group, and *Campylobacter*—correlating positively with the number of ETI treatment days, independent of other clinical variables (Supp. Fig. 2c). Conversely, levofloxacin inhalation was negatively associated with the relative abundance of several microbiome members, and it significantly reduced sputum diversity (Supp. Fig. 2c, e, Supplementary Data 5, 8). Higher sweat chloride levels correlated negatively with sputum Shannon diversity (metadeconfoundR: Spearman's rho = −0.37, FDR = 0.07), while the number of ETI treatment days positively correlated with both Shannon diversity (metadeconfoundR: Spearman's rho =0.34, FDR = 0.02) and the number of observed ASVs (metadeconfoundR: Spearman's rho=0.52, FDR < 0.001, Supp. Fig. 2e and Supplementary Data 8).

Our confounder-aware analysis shows that clinical improvements following ETI treatment, particularly reductions in sweat chloride levels, significantly influenced shifts in the sputum microbiome by reducing *Staphylococcus* abundance and promoting microbial diversity, while *Pseudomonas* abundance remained linked to host inflammation rather than ETI treatment duration.

**Direct antibacterial effects of ETI might further confer niche disadvantage to *Staphylococcus***

Despite physiological changes and daily medications affecting the microbiome, we hypothesized that ETI might exhibit direct antibacterial effects on the sputum microbiome. Accordingly, we tested the direct inhibitory effects of the ETI drugs on a selection of bacterial species commonly found in the lung microbiome. Elexacaftor and ivacaftor directly reduced the growth of several sputum microbiome members, including *Staphylococcus aureus*, but not *Pseudomonas aeruginosa* (Fig. 2g). Thus, in addition to the altered host physiology, the direct growth inhibitory properties of ETI might contribute to creating niche disadvantages for *Staphylococcus aureus* while sparing *Pseudomonas aeruginosa*. Interestingly, we observed an increase of several sputum microbiome genera (e.g., *Neisseria* and *Haemophilus*, Supp. Fig. 2c, d) which also had members that were not targeted by elexacaftor or ivacaftor in the bacteria-drug interaction testing. This underscores the potential niche-shaping impact of the modulators' antibacterial effects on lung commensals.

**Gradual shifts in gut microbiome composition in response to ETI treatment**

In contrast to the rapid changes observed in the respiratory tract microbiome, alterations in the gut microbiome emerged more gradually. No significant variance was detected within the first 3 months of treatment. However, notable shifts began to emerge between 6 to 12 months (PERMANOVA: Baseline vs. 6-12 months, $R^2 = 1.1\%$, $p = 0.001$), with further increased divergence from baseline seen at later time points (PERMANOVA: Baseline vs. 15-18 months and 21-24 months, $R^2 = 2.6\%$, $p = 0.001$). Unlike sputum samples, stool microbiomes continued to diverge over time, with significant differences detected between later time points (Fig. 3a, b).

In parallel, we observed a significant increase in microbial richness (number of observed ASVs) at later time points (15–18 months and

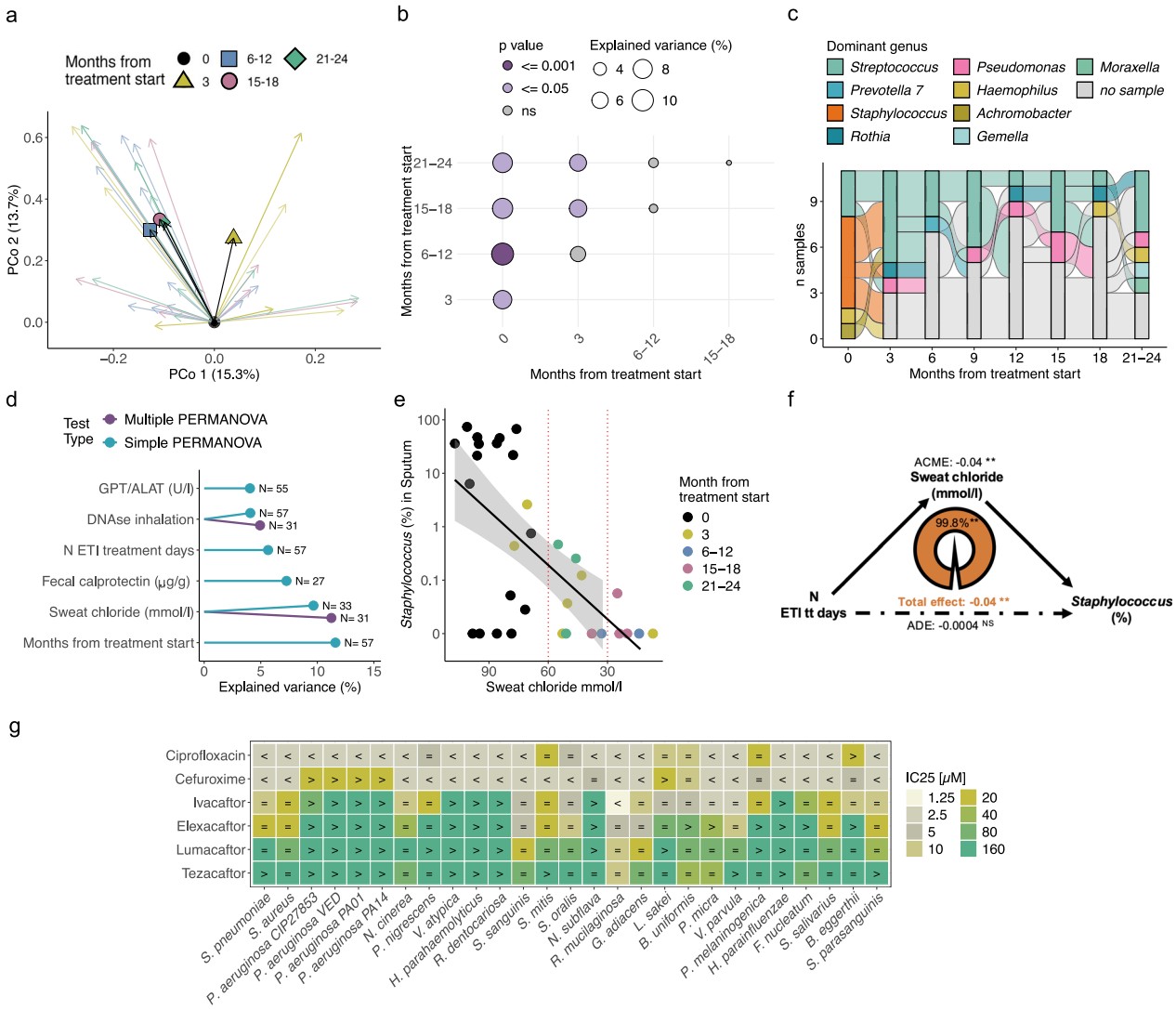

**Fig. 2 | ETI drives sputum microbiome into a new stable state via host and direct antibacterial effects. a** Sputum microbiome composition shifts from baseline (PERMANOVA on BC-dissimilarities, stratified by donor, adjusted for sex and age: $R^2 = 11.5\%$, $p = 0.001$). PCoA shows first two dimensions; arrows connect each sample to its baseline. Colors indicate months from ETI treatment start: black=baseline (N = 18), yellow=3 months (N = 8), blue=6–12 months (N = 13), red=15–18 months (N = 9), green=21–24 months (N = 9). Geometric markers = mean BC-dissimilarity from baseline per timepoint. **b** Microbiome variance ($R^2$) explained between sampling time points (PERMANOVA on BC-dissimilarities, stratified by donor). Circle size reflects $R^2$; color indicates p-value, details in Supplementary Data 17. **c** Alluvial plot illustrating *Staphylococcus* dominance in sputum microbiomes at baseline; colors denote the dominant genus per sample, edges connect samples from the same participant (N = 11). Missing samples are shown in gray. **d** Microbiome variance ($R^2$) explained by individual covariates. Blue: significant in simple, purple: in multiple PERMANOVA on BC-dissimilarities ($p < 0.05$), details in Supplementary Data 18. **e** *Staphylococcus* relative abundance correlates with CFTR

function (sweat chloride). Effect size=0.57, FDR = 0.015, metadeconfoundR. Sample time points are color-coded. Black line =linear fit; shaded areas = 95% confidence intervals. Red dashed lines show CF diagnostic thresholds: > 60 mmol/L = CF; 30-60 mmol/L = borderline; < 30 mmol/L = unlikely[55]. **f** Mediation analysis: The direct effect of N ETI treatment (tt) days on *Staphylococcus* relative abundance is not significant once the mediation effect is accounted for. **ADE** (Average Direct Effect): −0.0004 95%CI (−0.03, 0.03), $p = 0.998$. **ACME** (Average Causal Mediation Effect): −0.04, 95%CI (−0.07, −0.01), $p = 0.002^{**}$; **ADE** (Average Direct Effect): −0.0004 95% CI (−0.03, 0.03), $p = 0.998$; **Total effect:** Combined effect of ACME and ADE. −0.04 95%CI −0.06, −0.01, $p = 0.008^{**}$; **Mediation effect:** 0.998 95%CI (0.34, 3.33), $p = 0.010^{**}$. **g** CFTR modulators directly inhibit several lung microbiome members (e.g., Streptococcus, Staphylococcus) in vitro. Antibiotics were tested as controls. $IC_{25}$ = concentration for 25% growth inhibition after 22 h drug exposure compared to growth in 1% DMSO (i.e., untreated control). "<" = $IC_{25}$ below tested; ">" = $IC_{25}$ above tested dose.

21–24 months) compared to baseline (LME Estimates: 10.4 and 12.8, FDR = 0.029 and 0.015, respectively; Supp. Fig. 4b, and Supplementary Data 4). Notably, the presence of *Escherichia-Shigella* in stool decreased following ETI treatment (Fig. 3c, LME Baseline vs 6–12 months: Estimate = −0.39, FDR = 0.05, Baseline vs 15–18 months: Estimate = −0.5, FDR = 0.04).

To identify potential clinical drivers of the observed microbial shifts, we performed the two-step PERMANOVA analysis (Fig. 3d), as described previously. In the combined PERMANOVA (excluding the

variables pseudomonas_positivity_in_throat, CRP, and calprotectin due to insufficient data) four variables explained significantly variance: inhalative mannitol treatment ($R^2 = 1.7\%$, $p = 0.011$); months from treatment start ($R^2 = 1.6\%$, $p = 0.002$); the number of systemic antibiotic treatment days within the last 365 days before sampling ($R^2 = 1.5\%$, $p = 0.001$); and oral corticosteroid treatment ($R^2 = 1.3\%$, $p = 0.028$) (Fig. 3d).

In parallel, we found a negative correlation between the number of observed ASVs and the number of antibiotic treatment days

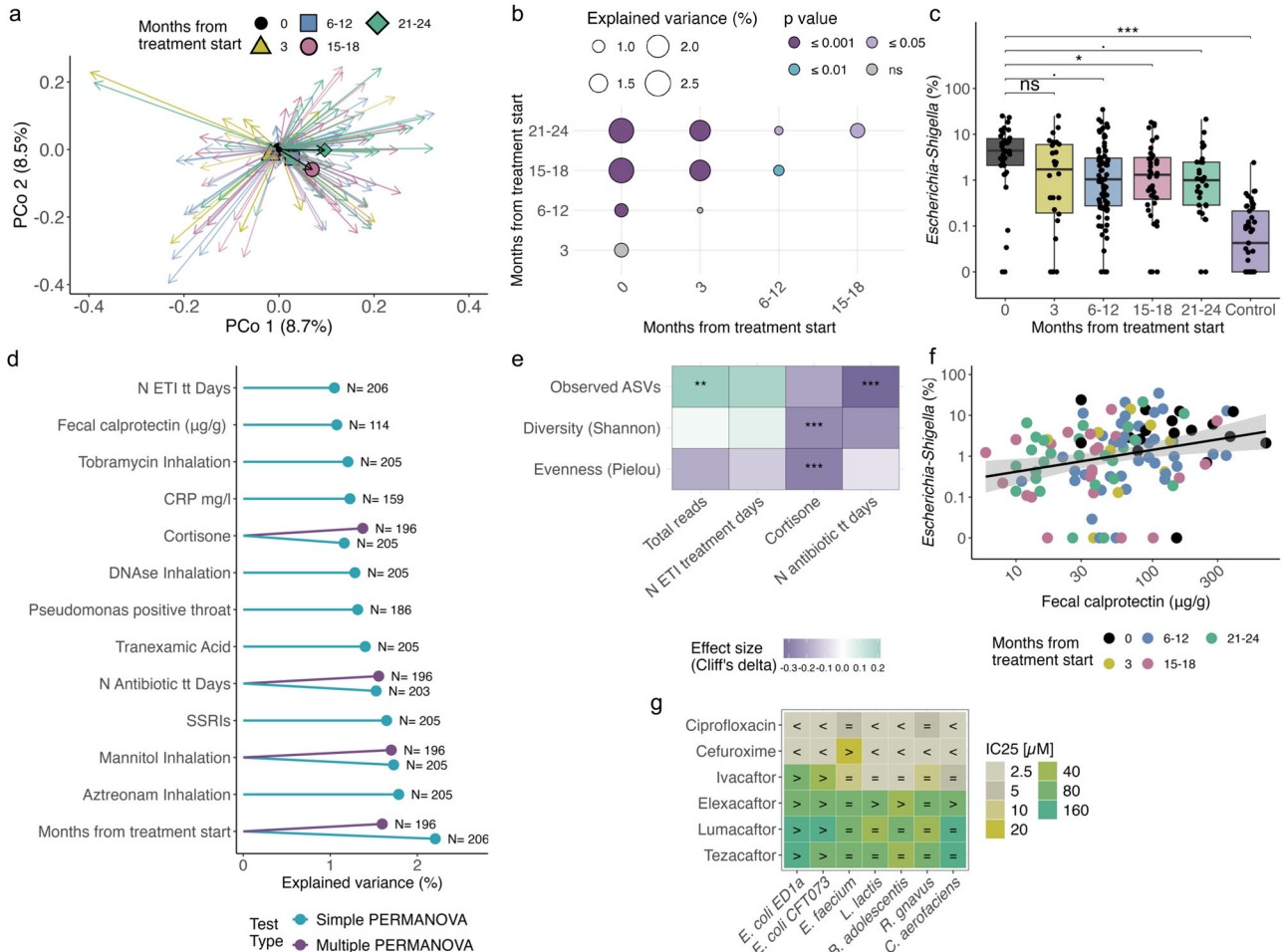

**Fig. 3 | Gradual shifts in gut microbiome composition in response to ETI are driven by antibiotics and steroids, including *Escherichia* decline linked to reduced intestinal inflammation. a** Stool microbiome composition shifts from baseline (PERMANOVA on BC-dissimilarities, stratified by donor, adjusted for sex and age: Months from ETI start $R^2$ = 2.3%, p = 0.001). PCoA shows first two dimensions; arrows connect each sample to its baseline. Colors indicate months from ETI treatment start: black=baseline (N = 34), yellow=3 months (N = 26), blue=6-12 months (N = 76), red=15-18 months (N = 40), green=21-24 months (N = 31). Geometric markers show mean shift from baseline. **b** Microbiome variance ($R^2$) explained between sampling time points (PERMANOVA on BC-dissimilarities, stratified by donor). Circle size reflects $R^2$; color indicates p-value, details in Supplementary Data 17. **c** Relative abundance of *Escherichia-Shigella* in stool samples across time and controls. Significance vs. baseline assessed by LME (ID as random factor). FDR < 0.1 (.), FDR < 0.05 (*), FDR < 0.01 (**), and FDR < 0.001 (***), values in Supplementary Data 7. Box plots show the median (line), IQR (box), 1.5× IQR range

(whiskers), and outliers (points beyond whiskers). Sample time points are color-coded. **d** Microbiome variance ($R^2$) explained by individual covariates. Blue: significant in simple, purple: in multiple PERMANOVA on BC-dissimilarities (p < 0.05), details in Supplementary Data 18. **e** Alpha diversity metrics (calculated on rarefied reads to minimum sequencing depth) significantly associated with clinical metadata. Heatmap hues represent signed effect sizes (Spearman's *rho* or Cliff's delta). FDR < 0.1 (.), FDR < 0.05 (*), FDR < 0.01 (**), and FDR < 0.001 (***). Features shown have effect size > |0.2|. Details in Supplementary Data 8. **f** Fecal calprotectin negatively correlates with *Escherichia-Shigella* abundance. Sample time points are color-coded. Effect size= 0.31, FDR = 0.04, without confounders reported by metadeconfoundR. line =linear fit; shaded areas = 95% confidence intervals. **g** CFTR modulators inhibit growth of several gut microbes in vitro. Antibiotics were tested as controls. IC$_{25}$ = concentration for 25% growth inhibition after 22 h drug exposure compared to growth in 1% DMSO (i.e., untreated control). "<" = IC$_{25}$ below tested; ">" = IC$_{25}$ above tested dose.

(metadeconfoundR: Spearman's rho = −0.33, FDR < 0.001), as well as a negative association between oral corticosteroid use and Shannon diversity (metadeconfoundR: Cliff's delta = −0.25, FDR < 0.001) (Fig. 3e).

The reduction in *Escherichia-Shigella* was positively correlated with decreases in fecal calprotectin, a marker of gastrointestinal inflammation (Fig. 3f, metadeconfoundR Spearman's rho = 0.31, FDR = 0.046). Further, *Escherichia-Shigella* relative abundance was higher in individuals that were taking proton pump inhibitors (metadeconfoundR Cliff's delta = 0.34, FDR < 0.001, Supp. Fig. 5c, d, and Supplementary Data 5), an association that has been found before[20] and further validates our findings.

Following up on this, we investigated whether the reduction of *Escherichia-Shigella* might be attributed to a direct antibacterial activity of any of the CFTR modulators and tested drug susceptibility in

our in vitro direct drug-bacteria interaction assay. *E. coli* was not directly affected by the modulators, although several commensal gut bacteria were (Fig. 3g).

Overall, shifts in the stool microbiome became apparent after 6 months of ETI, largely driven by antibiotic and corticosteroid use. The reduction of *Escherichia-Shigella* was linked to decreased intestinal inflammation, highlighting the potential health-promoting effects of these microbial changes.

**Throat microbiome: subtle but stable shifts with ETI treatment**
We observed distinct yet subtle shifts in throat microbiome composition from baseline to follow-up samples. Significant differences were found between baseline and follow-up sampling time points (PERMANOVA: Baseline vs. 3 months, $R^2$ = 2%, p = 0.037; Baseline vs. 6–12 months, $R^2$ = 1.4%, p = 0.014; Baseline vs. 15-18 months, $R^2$ = 2%,

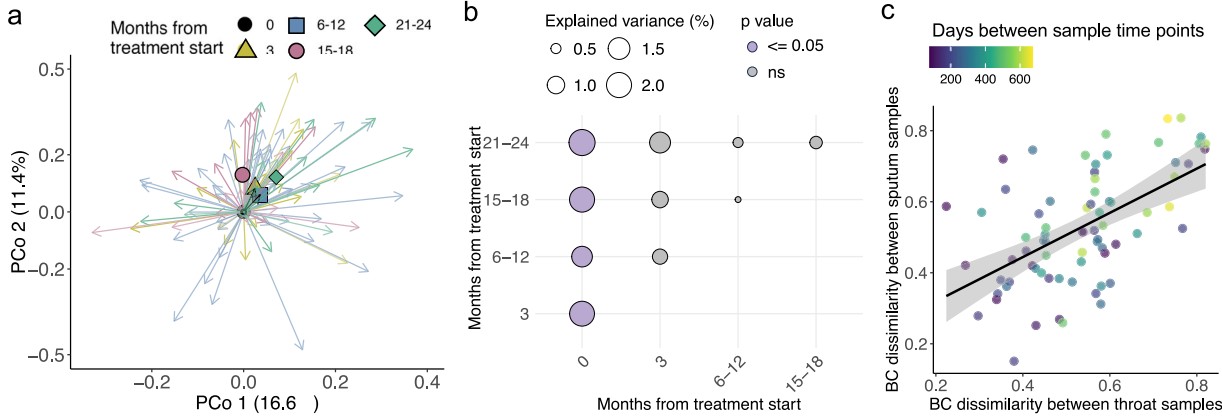

**Fig. 4 | Changes in throat composition are subtle and mirror to a certain extent changes in sputum microbiome. a** Throat microbiome composition shifts from baseline (PERMANOVA on BC-dissimilarities, stratified by donor, adjusted for sex and age: Months from ETI start $R^2 = 1.6\%$, $p = 0.018$). PCoA shows first two dimensions; arrows connect each sample to its baseline. Colors indicate months from ETI treatment start: black=baseline (N = 21), yellow=3 months (N = 29), blue=6–12 months (N = 80), red=15–18 months (N = 43), green=21-24 months (N = 40). Geometric markers show mean shift from baseline. **b** Microbiome variance ($R^2$) explained between sampling time points (PERMANOVA on BC-dissimilarities, stratified by donor). Circle size reflects $R^2$; color indicates p-value, details in Supplementary Data 17. **c** Each point represents a single participant, where the x-axis shows the BC-dissimilarities of throat-throat pairs, and the y-axis displays the BC-dissimilarities of sputum-sputum pairs for the same sampling interval. Point colors represent the sampling interval duration (in days), with purple indicating shorter intervals and yellow indicating longer intervals. Line shows linear fit; shaded areas indicate 95% confidence intervals. LME, controlled for ID-pairs and sample time point-pairs as random effects, revealed a strong positive association between throat-throat and sputum-sputum BC dissimilarities (LME Estimate = 0.51, $p = 1.81$e-05), suggesting that greater changes in the throat microbiome parallel greater shifts in the sputum microbiome. Additionally, sampling interval duration showed a weak but significant effect (LME Estimate = 2.12e-04, $p = 0.03$), indicating that longer intervals contribute to increased microbial dissimilarity.

$p = 0.047$; Baseline vs. 21–24 months, $R^2 = 2\%$, $p = 0.022$) (Fig. 4a, b). However, no significant differences were observed among follow-up samples, suggesting a shift to a "new stable state," similar to what was seen in sputum samples.

Unlike the sputum and stool microbiomes, we did not observe significant changes in taxonomic abundances in throat samples in response to ETI treatment. However, in a subset of paired sputum-throat samples (N = 90), we found that time point of sampling explained more variance than the sample type (simple PERMANOVA stratified by donor: $R^2$ sample type = 6.9%, p = 0.001; $R^2$ sample time point= 12.5%, p = 0.013, Fig. 5b). Furthermore, within-participant BC-dissimilarities over time in throat samples significantly correlated with those in sputum (Fig. 4c). This indicates that longitudinal changes in the throat microbiome paralleled shifts in the sputum microbiome, suggesting a degree of coordinated airway response to ETI therapy.

Importantly, this dynamic synchrony appeared unique to respiratory niches: we did not observe similar relationships between stool and throat or stool and sputum samples, suggesting distinct habitat-specific responses to treatment.

### Cross-habitat microbial colonization structure in pwCF
To further understand the cross-habitat microbial colonization patterns in pwCF, we analyzed the relatedness of microbial communities across throat, sputum, and stool samples (Fig. 5 and Supp. Fig. 7). PCoA on BC-dissimilarities revealed distinct clustering by sample type, explaining 26% of the variance ($p \le 0.001$; Fig. 5a, b, and Supplementary Data 16). While the time point of sampling did not affect gut-respiratory comparisons, it accounted for more variance in sputum-throat pairs than sample type (Fig. 5b, and Supplementary Data 16).

Sputum-throat pairs exhibited the highest intra-individual similarity, indicated by consistent Procrustes alignment and lower BC-dissimilarities (Fig. 5c, e). Conversely, stool-respiratory pairs were more dissimilar, reflecting distinct microbial communities (Fig. 5d, e). In paired sputum-throat samples, each site harbored approximately 70% of the ASVs detected in the other. In paired sputum-stool comparisons, sputum samples contained 13% of ASVs also present in stool, while stool samples shared 10% with sputum. Paired samples shared more ASVs across all site combinations than unpaired samples ($p < 0.001$; Fig. 5f, g), demonstrating measurable cross-habitat connectivity at the ASV level. Correlated alpha diversity metrics further linked the upper and lower airways (Supp. Fig. 7a).

Although *Pseudomonas* and *Staphylococcus* were not among the top 20 shared ASVs and showed poor abundance correlation between sites, their presence/absence was consistent and clinically informative, throat swabs predicted *Pseudomonas* in sputum with 87% sensitivity and 77% specificity (Supp. Fig. 7b, c, and Supplementary Data 15).

ETI treatment had minimal impact on gut-respiratory microbial structure, with no evidence of coordinated shifts across the gut-lung axis. In contrast, microbial profiles between throat and sputum samples remained closely aligned and were more influenced by treatment duration than sample type. These findings suggest localized, habitat-specific therapeutic effects and highlight the value of throat swabs as a minimally invasive proxy for sputum in CF airway monitoring.

### Comparison to healthy controls: CFTR modulator therapy shifts gut microbiome toward healthy control levels, reducing potentially pathogenic Proteobacteria
We compared the stool and throat microbiomes of our CF cohort with those of a sex and aged matched healthy control group. While differences in microbial composition between CF and controls were significant in both types of samples at all sampling time points, the variance explained was notably greater in stool (approximately 10%) compared to throat (around 3%) (Fig. 6a).

In addition, healthy controls had higher Shannon diversity in stool samples compared to any CF sampling time point (e.g., LME Control vs CF Baseline: Estimate = −0.6, FDR < 0.001, Supp. Fig. 5a and Supplementary Data 9). However, this pattern was not observed in throat samples, where diversity levels between pwCF and controls were more similar (Supp. Fig. 8a, b). Wilcoxon tests revealed minimal taxonomic differences in throat samples, while stool samples consistently showed many taxa significantly differing between CF and controls (Fig. 6b, Supp. Fig. 8a,b and Supplementary Data 10).

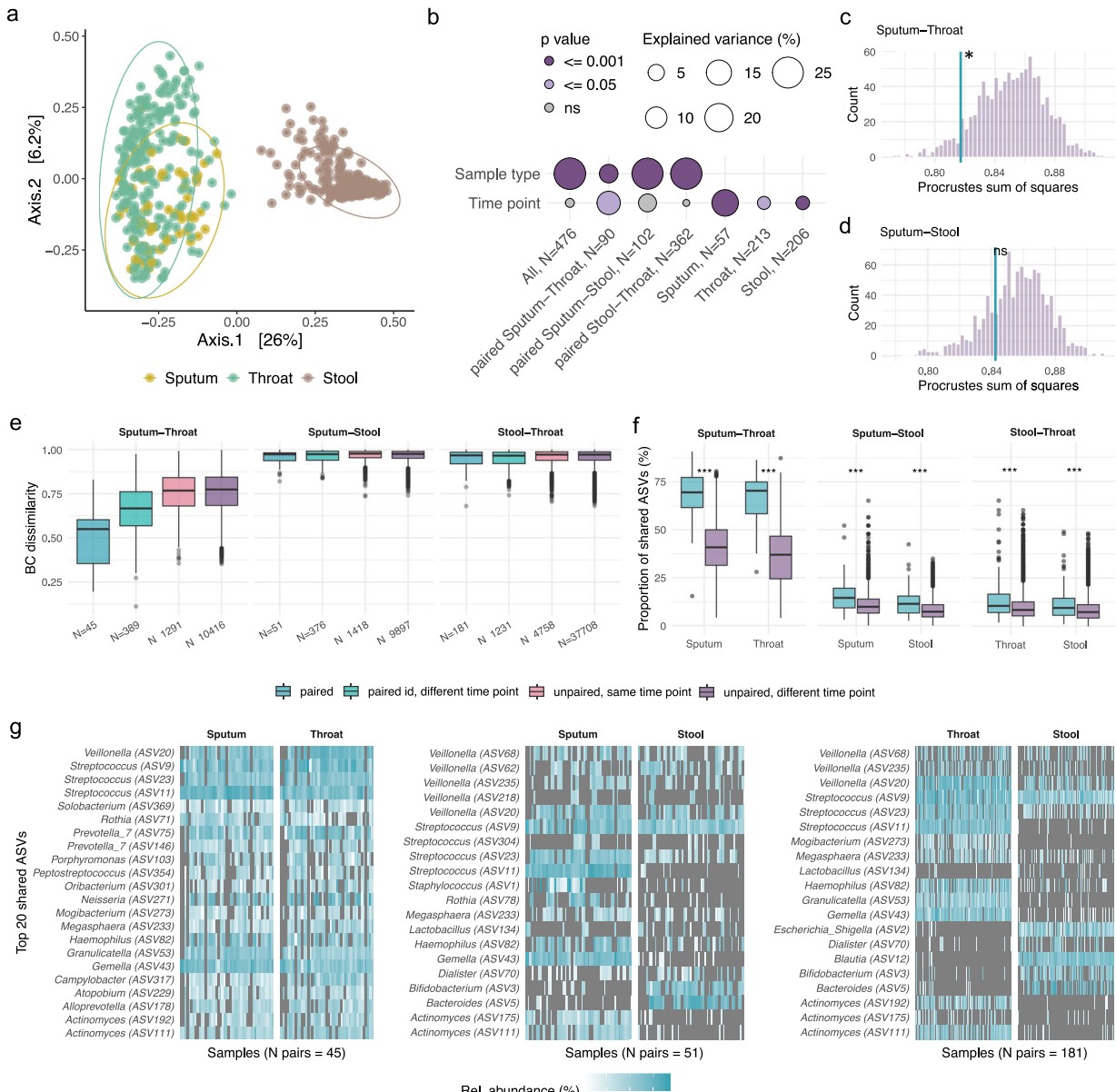

**Fig. 5 | Cross-habitat microbial colonization structure in pwCF. a** PCoA of BC-dissimilarities across sputum, throat, and stool samples from pwCF. Samples cluster by type, explaining 25% of the total variance (PERMANOVA stratified by participant ID and adjusted for Months since treatment start; $p \leq 0.001$). **b** ID-stratified simple PERMANOVA showing variance in microbial composition explained by sample type and sample time point (Visit 1-9) across habitat combinations (details in Supplementary Data 16). **c, d** Procrustes analysis comparing microbial community structure between habitats. Turquoise line: Procrustes residual ($m^2$) for paired samples; purple histogram: null distribution from 1,000 permutations of randomly matched unpaired samples (sputum–throat $N = 45$, sputum–stool $N = 51$; $p < 0.05$; *ns* not significant. **e** Pairwise BC-dissimilarities between habitats, stratified by sample relatedness: paired (same participant and timepoint), same ID, different timepoint (same participant at different timepoints), unpaired, same timepoint (different participants sampled at the same timepoint), unpaired, different timepoint (different participants at different timepoints). **f** Proportion of ASVs shared between habitats, shown per site and stratified by sample relatedness. **g** Heatmaps show the relative abundance (log-scaled) of the top 20 shared ASVs across paired samples: sputum-throat (left), sputum-stool (middle), and throat-stool (right). Each column represents one sample, grouped by sample type, and each row corresponds to a shared ASV labeled with its taxonomic assignment and ASV ID. Color intensity represents relative abundance, with grey indicating absence. The top 20 ASVs were found in 64.4% of sputum-throat pairs (29 of 45), 13.7% of sputum-stool pairs (7 of 51), or 10.5% of throat-stool pairs (19 of 181). Box plots show the median (line), IQR (box), 1.5× IQR range (whiskers), and outliers (points beyond whiskers).

We investigated whether ETI treatment shifted CF microbiomes closer to those of healthy controls. In stool samples, several taxa showed transitions toward control-like abundances. Using an IQR-Z-score transformation (where 0 indicates no deviation from the control median), we analyzed 52 taxa significantly different at baseline to controls and tested if their Z-scores converged toward controls over time. 13 taxa exhibited significant changes, with 11 showing trends toward normalization. Proteobacteria, including Gammaproteobacteria, Enterobacterales, *Enterobacteriaceae*, and *Escherichia/Shigella*, *Oscillospiraceae* and the *Clostridium innocuum* group Z-scores declined, while Firmicutes, Oscillospirales, *Ruminococcaceae*, and *Romboutsia* Z-scores increased, aligning closer to controls (Fig. 6c–f, and Supplementary Data 11). In throat samples, normalization to control levels were mostly seen in the Actinobacteriota phylum, specifically Actinobacteria, Micrococcales, and *Rothia* (Supp. Fig. 8b, c, and Supplementary Data 11).

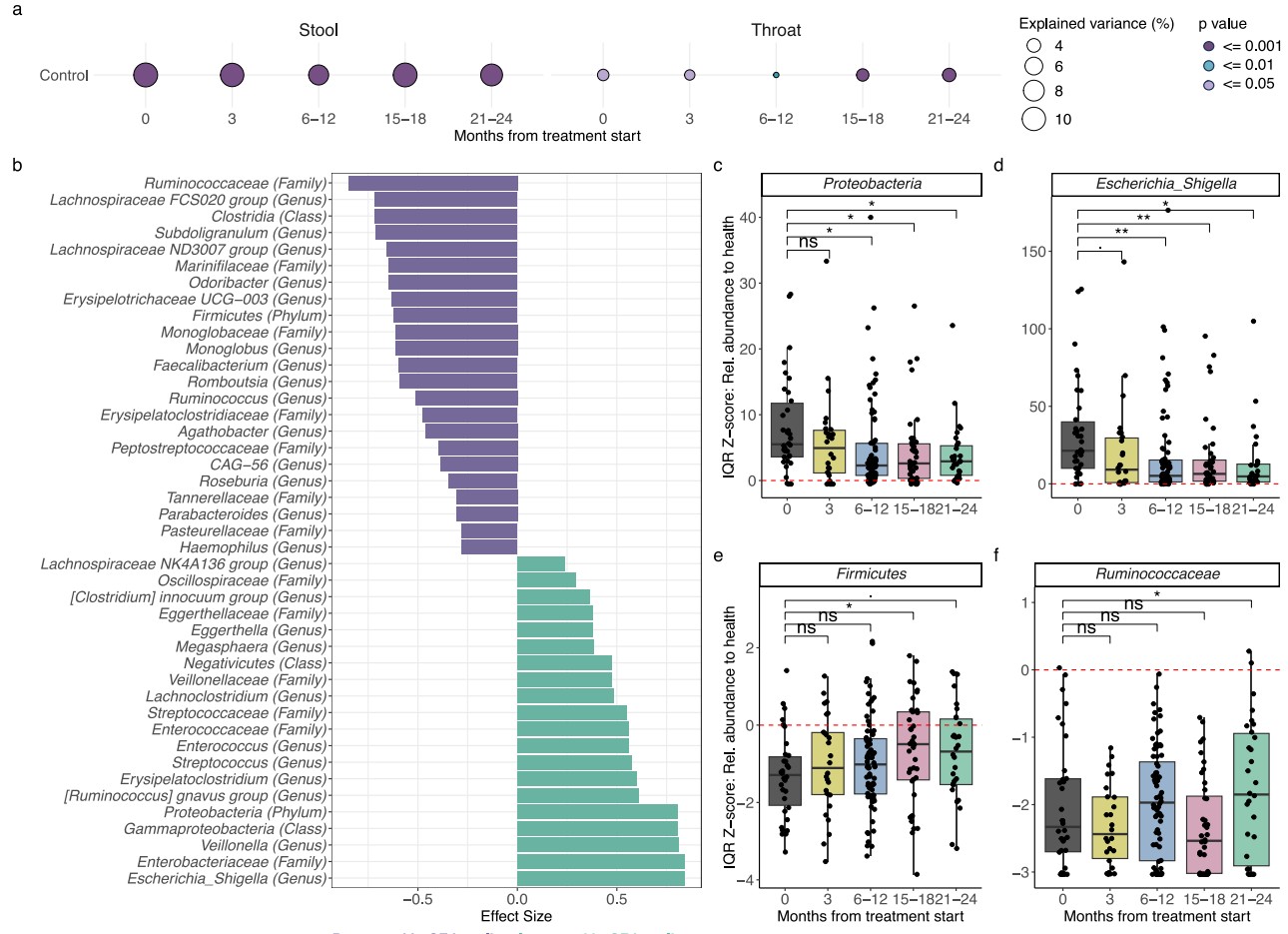

**Fig. 6 | Comparison between CF and healthy control throat and stool microbiomes. a** Bubble plot showing microbiome variance ($R^2$) between healthy controls and each CF timepoint (PERMANOVA on BC-dissimilarities). Circle size reflects $R^2$, color indicates significance (p-value), details in Supplementary Data 17. Left: stool, right: throat. Stool samples show markedly higher variance than throat, indicating stronger microbiome differences between pwCF and healthy controls. **b** Bar plot displaying taxa significantly differentially abundant between healthy controls and CF baseline stool samples (Wilcoxon test, FDR < 0.05). Positive effect sizes (green): taxa enriched in CF baseline samples; negative effect size (violet): reduced in CF baseline samples compared to healthy controls. **c–f** Example taxa for which IQR-Z

Score changes over ETI treatment time in CF stool samples and align towards abundances observed in healthy controls. Boxplots depict the distribution of IQR-Z scores for microbiome taxa across different time points following ETI treatment initiation (Baseline, 3 months, 6–12 months, etc.). Box plots show the median (line), IQR (box) and 1.5× IQR range (whiskers). Z-score of 0 indicates no deviation from the control median, and is depicted as a dashed red horizontal line. Dots: individual samples. Significance of changes compared to baseline (0 months) is determined using a linear mixed-effects model (lme), adjusted for participant ID as a random factor. FDR < 0.1 (.), FDR < 0.05 (*), FDR < 0.01 (**), and FDR < 0.001 (***), exact p- and FDR values in Supplementary Data 7.

---

These findings suggest that CFTR modulator therapy fosters a shift in the gut microbiome of pwCF towards a composition more reminiscent of healthy individuals, particularly in key microbial taxa.

## Discussion

In this study, we demonstrate how 24 months of ETI modulator therapy, through induced physiological changes and direct antibacterial effects, restructures the CF microbiome across three distinct body habitats (upper and lower respiratory tracts and the gastrointestinal tract). To the best of our knowledge, this is the first study to investigate these anatomically distinct microbiomes in CF simultaneously. While previous studies have primarily described microbial compositional changes associated with ETI therapy, we set out to uncover the host-associated alterations driving these changes. Using confounder-aware statistical methods, we delineated host-microbiome interactions in this complex clinical context characterized by multiple morbidities and treatment modalities.

By accounting for clinical confounders, we disentangled the contributions of ETI treatment from those of reduced antibiotic use on microbial composition. While other studies have reported a link between lower lung function and reduced microbial alpha-diversity[6,21], our findings indicate that reduced sputum alpha-diversity is more strongly associated with inhaled levofloxacin treatment, than with lung function. Deconfounding these relationships is critical, as increased microbiome diversity observed in pwCF on ETI could be driven by reduced antibiotic exposure and not solely by CFTR modulation. In addition, response to ETI may be site-specific. Indeed, in our deconfounded analysis, sputum microbiome diversity was positively associated with ETI treatment duration, while stool diversity was more strongly related with reduced systemic antibiotic use.

Multiple cohort studies have reported a general decline in *Staphylococcus* and *Pseudomonas* sputum burden following ETI initiation[8,16,17]. However, many patients remain colonized by the same pathogens post-treatment. Interestingly, one study observed no significant change in either organism and even noted an increase in *Pseudomonas* detection by culture[7]. These findings point to host- and pathogen (strain)-specific factors as likely contributors to the variable response to ETI therapy. Despite the modest follow-up sample size, sputum samples from the studied population clearly showed a statistically significant (FDR < 0.05) reduction in *Staphylococcus* following

ETI therapy. We note that sputum producers represent a subset of pwCF who differ clinically from non-producers (Table 2), and therefore this subgroup may not fully capture the spectrum of lung microbiome treatment responses. However, CFTR modulator therapy clearly changes the airway microbial colonization landscape. As a result, *Staphylococcus* fitness advantages, such as salt tolerance[22], relevant in the context of elevated salt concentrations in CF sputum[23], immune evasion and antimicrobial resistance determinants[24], no longer offer this genus a fitness advantage in the context of ETI treatment. Moreover, restoration of CFTR function in neutrophils and macrophages[25,26] likely contribute to improved clearance of *Staphylococcus*. Our study revealed that reduced S*taphylococcus* abundance was mediated by improved CFTR-function (based on reduced sweat chloride levels) and the direct antibacterial effects of two of the three components (Elexacaftor, Ivacaftor) of this CFTR modulator therapy. Thus, ETI treatment appears to render the sputum microenvironment less favorable to *Staphylococcus* colonization. However we note in our study that the bacterial burden in the airways does not change pre- or post-ETI treatment initiation. Instead, we observed that *Staphylococcus* is replaced either by commensal airway microbiome members or by respiratory pathogenic genera including *Haemophilus* and *Pseudomonas*. Thus, the clinical course during ETI treatment, and in particular susceptibility to pulmonary exacerbation, may be dictated by the dominant airway microbiome that arises post *Staphylococcus* depletion.

On the other hand, *Pseudomonas* persisted despite ETI therapy and associated improvements in host pathophysiology. While systemic IgG levels (a marker of chronic inflammation), decreased, plasma leukocyte counts remained unchanged specifically in *Pseudomonas*-positive participants. Additionally, *Pseudomonas* abundance in sputum correlated with leukocyte counts, highlighting its well-known pro-inflammatory properties in CF airways[27]. Unlike *Staphylococcus*, ETI active agents had no direct antibacterial effects on *Pseudomonas aeruginosa*. While some studies report a decline in *Pseudomonas* abundance, most found that it persisted in the majority of ETI treated pwCF, with the same strains often detected pre- and post-treatment initiation[28]. This persistence may, in part, reflect cohort-specific differences; for instance, our sputum follow-up cohort included a greater proportion of older patients with more advanced lung disease, which are factors previously associated with reduced pathogen clearance[17]. Beyond resistance to the active agents of ETI, *P. aeruginosa* displays impressive genomic plasticity and capacity for adaptation in the airways[21,29]. Populations of *P. aeruginosa* are known to rapidly evolve new variants even in the ETI environment, likely adapting to the altered selective pressures in CFTR-corrected airways[30]. Our findings emphasize the resilience of *Pseudomonas*, its role in sustaining inflammation, and the challenges it poses in CF care.

In our study, we demonstrate a strong link between intestinal inflammation and gut microbiome composition, specifically the relative abundance of *Escherichia-Shigella* in the context of ETI administration. Consistent with prior studies[31], we observed elevated *Proteobacteria* in the CF gut compared to healthy controls, driven primarily by *Escherichia-Shigella*. Following ETI treatment, *Proteobacteria* levels significantly declined, with *Escherichia-Shigella* abundance trending toward levels observed in healthy control subjects. Notably, reduced *Escherichia-Shigella* relative abundance correlated with decreased fecal calprotectin, aligning with prior findings linking *Enterobacteriaceae* to intestinal inflammation under conditions of CFTR modulation with ivacaftor[32]. *Escherichia*, as a facultative anaerobe, thrives in inflammatory environments by tolerating higher oxygen, using respiratory electron donors and epithelial-derived nutrients like ethanolamine[33]. Further, *Escherichia* has a high capacity to resist host-derived antimicrobials and antibiotics. Therefore, reduced inflammation and antibiotic use during ETI likely disadvantaged *Escherichia*, facilitating microbiome recovery.

Given *Escherichia's* adaptation to inflammatory environments and antibiotic resistance, reduced inflammation and antibiotic use during ETI likely disadvantaged them, facilitating microbiome recovery. This reciprocal relationship between intestinal inflammation and *Enterobacteriaceae*, demonstrated in mouse models[34] and studies on Adherent-Invasive *Escherichia coli* (AIEC)[35], underscores the active modulation of both microbiome niche and host physiology by ETI. Such modulation of the gut microbiome is clinically significant, as early gut microbiome health in pwCF has been shown to be a stronger predictor of growth outcomes in children with CF than modulator use alone[36].

Throat microbiome changes with ETI treatment were minimal, aligning with prior studies[37]. However, reduced sputum production with ETI complicates diagnostics[17], highlighting the clinical importance of exploring the throat-lung connection. In our study, strong intra-individual correlations between throat and sputum microbiome shifts emphasize their relationship, warranting further study, especially given conflicting reports on the reliability of throat swabs for assessing lung infections[38].

Despite the challenges of in vivo human studies, our comprehensive statistical approach is a major strength, accounting for confounders such as antibiotic use, polypharmacy, inflammation, and participant-specific factors. Our findings align with prior mouse models and clinical CF studies, reinforcing their validity and highlighting host-microbiome dynamics. However, reliance on 16S rRNA sequencing limits resolution to the genus level, restricting insights into functional repertoire and genetic adaptations. Further, CFTR modulator concentrations in sputum remain uncertain[39], with sub-therapeutic levels possibly interacting with antibiotics to disadvantage pathogens, as seen in animal studies[15]. In vitro models also lack key host factors, microbial interactions, and in vivo drug concentration accuracy. Although our sample size is modest, the study design allows robust statistical inference within the studied population, including FDR control (< 0.05), which minimizes the risk of spurious findings. Nonetheless, the single-center design and smaller subgroup sizes limit generalizability to populations with different clinical characteristics.

Larger, multi-center cohorts with extended follow-up and integrated metagenomic, multi-omic, and longitudinal designs will be critical to validate and expand upon these findings. Given the heterogeneity of CF and especially the CF lung microbiome, such efforts will be particularly valuable for capturing the full spectrum of sub-presentations and populations (e.g., *Pseudomonas*- or *Staphylococcus*-colonized vs. non-colonized, sputum producers vs. non-producers), and for disentangling the diverse factors shaping host–microbiome interactions in this disease. Moreover, while our study includes up to 24 months of follow-up, it remains possible that later changes or further normalization toward healthy states may emerge only with longer-term observation, providing an exciting opportunity for future investigation. Despite these limitations, our study, through multi-site (respiratory and gut) sampling, a comparatively large matched healthy control group, and a statistical framework accounting for confounding and mediation, provides essential groundwork for future large-scale investigations. Overall, this work represents a critical step in understanding how ETI therapy modulates microbiome–host interactions in CF across different habitats and highlights the potential for reversing dysbiosis through improved host physiology.

## Methods
### Participants and ethics
The study recruited participants from the pediatric pulmonology outpatient clinic of the University Medical Center Mainz (UMCM), which, in accordance with standard CF care in Germany, provides continuous care for both pediatric and adult patients within the same specialized center. Individuals, and in the case of minors, their legal

guardians or parents, provided informed consent before inclusion. Approval for the study design was obtained from the local ethics committee (Landesärztekammer Rheinland-Pfalz 2020-15541), and participant recruitment adhered to the principles of the Declaration of Helsinki. Additionally, the study was registered in the German Clinical Trials Register under the identifier DRKS00023862. Participants with a confirmed CF diagnosis (heterozygous or homozygous for F508del) and untreated with ETI before, were recruited between October 2020 and June 2022. Initially limited to individuals aged 12 years and older, the inclusion criteria expanded to encompass children 6 years and older following the approval of ETI for this age group during the study. Healthy age and sex-matched control participants were recruited between December 2021 and January 2023. They were recruited from the general population through local advertisements in our children's hospital and community outreach. They were matched with the pwCF cohort based on age and sex to ensure comparability. Healthy controls were required to be non-medicated (except for hormonal contraceptives), free from chronic illnesses, and without antibiotic use or signs of infection in the 4 weeks preceding recruitment. Due to ethical and practical constraints we could not obtain sputum samples from healthy controls.

### Study design

The phase IV study aimed to evaluate clinical and microbiome responses to ETI in previously ETI-untreated pwCF. Over a 24-month period, we longitudinally sampled sputum, throat, and stool specimens every three months from a cohort of 35 pwCF. Concurrently, we collected clinical metadata, including medication details, lung function, sweat chloride levels, and inflammatory markers in serum and stool. Healthy controls submitted a one-time stool and throat sample along with essential health details, including any antibiotic treatment in the past 12 months and dietary habits.

### Collection of clinical data and samples

Stool samples were collected by participants at home using kits designed to protect them from light and oxygen, as detailed in ref. 40. Samples were refrigerated at 4 °C for up to 24 hours before being frozen at −80 °C for processing, following the protocol outlined in ref. 40. Deep throat swabs, collected by a trained nurse using sterile cotton swabs, and spontaneous sputum samples, both gathered in sterile containers, were immediately frozen at −80 °C for further processing. At baseline, visits were delayed if participants had taken antibiotics within the last 2 weeks; however, for all other time points, antibiotic intake was not an exclusion criterion, though recent antibiotic use was accounted for in the statistical analysis.

Sweat chloride levels were assessed at baseline and at 3, 12, 18, and 24 months post-treatment initiation using standard procedures, employing Pilocarpine iontophoresis. In cases of missed scheduled measurements, additional sweat chloride assessments were conducted at alternative time points. Routine check-ups included the collection of blood samples through standard clinical procedures, sent to the UMCM central laboratory for comprehensive measurements covering blood count, glycated hemoglobin (HbA1c), liver enzymes, and inflammatory parameters. Calprotectin in stool was measured at the UMCM central laboratory using a monoclonal antibody with the Quantum Blue® fecal extended assay (BUHLMANN Diagnostics Corp; Amherst, USA).

Lung function, determined by ppFEV1 (predicted percent of forced expiratory volume in one second) and ppFVC (predicted percent of forced vital capacity), was assessed using body plethysmography with the Vyntus Body equipment from Vyaire®. Standard procedures outlined in the European Respiratory Society Technical Statement, as described by ref. 41, were strictly followed.

All participants were requested to disclose their basic dietary preferences, categorized as omnivore, lacto-ovo-vegetarian, pescatarian, vegan, gluten-free, paleo, or ketogenic. Additionally, participants provided information on stool consistency based on the Bristol stool scale[42].

All analyses were adjusted for birth-assigned sex as recorded in the medical records to account for potential sex-related differences in disease presentation and treatment response.

### Microbiome sequencing

The sputum samples were treated with Benzonase to deplete extracellular human DNA after collection[43]. DNA extraction from all samples (sputum, throat swabs, stool) was performed using DNeasy PowerSoil Pro Kits (QIAGEN, Germany) following the manufacturer's instructions. Subsequently, cell lysis was carried out by bead-beating at 30 Hertz for 2 cycles of 7 minutes each on a TissueLyser (QIAGEN, Germany).

The V4 region of the 16S rRNA gene was amplified using the primers 515 f and 806r (forward: 5′-GTGCCAGCMGCCGCGGTAA-3′, reverse: 5′-GACTACHVGGGTWTCTAATCC-3′), and indexes were added using the Nextera XT DNA Library Preparation Kit (Illumina, Inc., USA). Sequencing was performed on a MiSeq platform (Illumina, Inc., USA). Technical metadata were recorded, including the extraction date, material, and DNA quantity after extraction. In addition to the participant samples, negative controls were also sequenced for quality control. Amplicon Sequence Variants (ASVs) were determined from the raw sequencing data, and their corresponding reads were assigned. The taxonomy of individual ASVs was determined using DADA2[44] and the SILVA database[45].

### 16S marker gene qPCR for bacterial load quantification

Bacterial load in sputum samples was quantified using 16S marker gene qPCR on an Applied Biosystems QuantStudio 3 system. The reaction, conducted in 96-well plates with SYBR-Green, consisted of duplicate amplifications in a 15 μL volume, including 7,5 μL 2x SYBR Green PCR Master Mix, 500 nM of each primer, and 1.2 μL template DNA (0.5 μg/μL). The standard protocol involved an initial cycle at 95 °C for 10 min, followed by 40 cycles at 95 °C for 15 s and 1 min at 60 °C. Melting curve analysis ensured specificity. Standard curves for quantification used 10-fold serial dilutions ranging from 10^8 to 10^0 copies of the E. coli 16S rRNA gene.

### Direct drug-bacteria interaction testing

**Selection of bacterial strains and growth media.** To enable a broad overview of the direct effect of CFTR modulators on bacterial members of the lung microbiome, 24 bacterial species considered part of a CF and/or healthy lung microbiome (Supplementary Data 12) were selected from our strain collection. Additionally, gut bacterial species were selected for in vitro drug testing if they belonged to genera that either changed in abundance following ETI treatment in patients or showed significant associations with clinical metadata. Growth of bacterial species from the lung microbiome was assessed in Brain Heart Infusion (BHI) medium and two of its variants (BHI + + and LYBHI), as follows: For BHI preparation, 37 g of BHI broth powder (Thermo Fischer Scientific (Cat. No: CM1135B); Germany) was dissolved in 1000 mL distilled water (dH2O) and mixed using a magnetic stirrer. Sterilization was achieved by autoclaving at 121 °C for 15 minutes. For BHI + + , 10 mL of Hemin solution (50 mg Hemin and 1 mL 1 M NaOH filled up with dH2O to 100 mL; filter sterilized) and 15 mg NAD were added after the medium was autoclaved and cooled to room temperature. LYBHI was prepared by dissolving 37 g of BHI broth powder with 5 g of yeast extract, 0.5 g cellobiose, 0.5 g maltose and 0.5 g cysteine in 1000 mL dH2O using a magnetic stirrer. 15 minutes autoclaving at 121 °C sterilized the medium. Growth of gut bacterial species was assessed using modified Gifu anaerobic broth (mGAM). To prepare the medium, 41.7 g of mGAM powder (Nissui Pharma Solutions, Cat. No: 05426-GMM-0300) was dissolved in 1000 mL of distilled water (dH2O), mixed with a magnetic stirrer, and autoclaved at

115 °C for 15 minutes. All media were stored at 4 °C when not in use and, for anaerobic and microaerophilic growth assays, brought into the anaerobic chamber at least one day before usage to deplete/reduce oxygen.

**Drug plate preparation.** First, 70 mM stocks of all four CFTR modulators (see *Drugs and solvent included in the screen:* Supplementary Data 13) and 20 mM (ciprofloxacin) or 50 mM (cefuroxime) stocks for the antibiotics (all stocks in DMSO) were created. Out of these stocks, the highest 100-fold working concentration solution of all tested drugs −16 mM for the CFTR modulators and 2 mM for the antibiotics - was prepared in DMSO.

The assay plates containing a 2-fold working concentration were prepared and the IC25 screen was conducted as described in a recently published protocol[46]. In short, first a master drug plate containing the 100-fold working concentration of each tested drug in DMSO (i.e., from 16 mM to 0.125 mM in 2-fold steps for the CFTR modulators and from 2 mM to 0.25 mM in 2-fold steps for the antibiotics) was created. Using a pipetting robot (Eppendorf, epMotion96 (Serial number: 506911402082); Germany), 15 µL of each well of the master drug plate was transferred into a 96-well deep well plate and mixed with 735 µl of BHI, BHI++ or LYBHI, respectively. 50 µL per well of the resulting 2-fold working concentration solution was then distributed into 96-well u-bottom ready-to-use plates (Thermo Fischer Scientific (Cat. No: 168136); Germany) that were sealed with aluminum foil (Beckman Coulter (Cat. No: 538619); Germany) and stored at −20 °C prior to usage. All plates were stored for four weeks at most and, for anaerobic or microaerophilic experiments, brought into the chamber one day before the experiment to deplete/reduce oxygen.

**IC25 determination.** All lung bacterial strains were first grown overnight at 37 °C on BHI plates and then for another overnight culture in 5 mL liquid BHI, BHI++ or LYBHI under anaerobic conditions (Coy Laboratory Products Inc.), microaerophilic conditions (i.e., 5 vol-% oxygen, Coy Laboratory Products Inc.) or aerobic conditions, respectively. Gut bacterial strains were grown anaerobically at 37 °C (2 vol-% $H_2$, 12 vol-% $CO_2$, 86 vol-% $N_2$, Coy Laboratory Products Inc.). Initially, they were cultured on mGAM agar plates for one overnight incubation, except for *Ruminococcus gnavus* and *Bifidobacterium adolescentis*, which required two overnights. Colonies were then inoculated into 5 mL liquid mGAM and incubated overnight at 37 °C. Then, both lung and gut bacterial Then, cultures were diluted to an OD600 of 0.02 and aliquots of 50 µL were added to the wells of the ready-to-use plates containing 50 µL of the test compound solution at twice the final concentration, whereby two biological replicates were tested on one plate. Final tested drug concentration were 160 µM to 1.25 µM for CFTR modulators and 20 µM to 2.5 µM for antibiotics. Plates were sealed with a breathable membrane (Sigma-Alrich via Merck (Cat. No: Z380059-1PAK); Germany) and bacterial growth at 37 °C was monitored for 22 h using hourly OD measurements in a BioTek Epoch 2 microplate reader which was placed within the anaerobic chamber (Agilent (Serial number: 19031316); Germany). The experiment was conducted in at least two independent biological replicates for all bacteria and growth curves were analyzed using the R package *NeckaR* (https://github.com/Lisa-Maier-Lab/neckaR) for IC25 values. The experimental protocol for direct drug-bacteria interaction testing has been described in full detail in ref. 46.

**Statistical analysis**
All statistical analyses were performed in the R environment (v4.1), using the phyloseq[47] package for sequencing and clinical data management. All analyses were conducted with the "na.action = exclude" parameter, ensuring that only available data points for each participant were used (e.g., if a participant provided five throat samples across the study, all five were included in the analysis). No imputation

of missing values was performed. All annotated data, metadata, and analysis scripts are available at (https://github.com/RebeccaLuise/IMMProveCF_public) with raw sequences accessible at (http://www.ncbi.nlm.nih.gov/bioproject/1080555) (PRJNA1080555).

**Reproducibility.** Sample size was predetermined using power calculations based on paired t-tests, indicating 80% power to detect medium effects at N = 35. Samples failing sequencing quality control were excluded; if no biosample remained for a given timepoint, associated clinical data were also excluded. Experiments were not randomized, and investigators were not blinded during sample processing or outcome assessment.

**Sequence data quality assessment, count transformation and taxa filtering.** To ensure adequate sequencing depth we performed rarefaction analyses and assessed the coverage of our sequencing data, as well as the distribution read counts across sample types. Following this, we excluded samples with a total read count < 5000 reads (N = 19).

Alpha diversity was calculated on both raw and rarefied counts (rarefied to the minimum sampling depth per sample type) using the alpha function from the microbiome package[48], results are summarized in Supplementary Data 14. For multivariate analysis (beta diversity: BC dissimilarities) and taxonomic differential abundance testing, sequencing data was normalized using Total Sum Scaling (TSS).

For taxonomic differential abundance testing, ASVs were prefiltered at a minimum prevalence of 30% (e.g., 76/252 samples) or 10% (6/57) for sputum samples and aggregated to at least the genus level using tax_glom (phyloseq). Confounder-aware analyses with MetadeconfoundR[19] applied a 25% prevalence threshold across all sample types.

**Multivariate analysis of microbiome composition.** Bray-Curtis (BC) dissimilarities were computed using the unweighted distance function of the phyloseq package. A stratified permutational multivariate analysis of variance (PERMANOVA) was conducted using the adonis2 function from the vegan package[49]. To control for individual-level variation, the dataset was stratified by participant ID and analyzed for marginal effects of variables tested. The model applied was: BC dissimilarities ~ tested_variable(s) + Blocks(ID) + ϵ (error term). Permutation tests (999 iterations) under a reduced model were performed to compare observed data with randomly permuted datasets.

A two-step PERMANOVA approach was applied. First, in a simple PERMANOVA, each variable listed in Supplementary Data 6 was independently tested to screen for significant contributors to microbial community variation (two-sided $p < 0.05$). Variables identified as significant in this step were incorporated into a multiple PERMANOVA model to assess their joint influence while controlling for confounding effects. To address collinearity, Variance Inflation Factors (VIF) were calculated, and variables with high collinearity (VIF ≈ 5) with ETI treatment time were excluded. Additionally, only samples with complete data for the tested variables were included, reducing the dataset size in the multiple PERMANOVA.

**Univariate analysis of clinical parameters, alpha diversity and differential taxa abundance.** To analyze the evolution of clinical parameters and alpha diversity measures across sampling time points or in comparison to healthy control samples, we employed linear mixed-effects models (LMEs). These were calculated using the lmerTest package in R[50]. Sex and age were included as fixed effects to account for their potential influence on the dependent variable, while participant ID was added as a random factor to account for repeated measures. Two-sided p-values from the models were corrected for multiple testing using the false discovery rate (FDR) (Benjamini-Hochberg procedure[51]).

To identify significant deviations in relative taxonomic abundances from baseline or healthy control samples we employed Wilcoxon signed-rank test. To account for the distinction between paired and non-paired samples, we next tested the rank-transformed feature counts against the sample time point with ID as a random factor in a linear model. Two-sided p-values were adjusted for multiple comparisons at each taxonomic level using FDR. Results across all time points and taxonomic levels were combined for visualization, highlighting effect sizes and significance levels (FDR).

A confounder aware statistical testing tool (MetadeconfoundR[19]) was used to test associations between relative taxonomic abundances and the clinical and technical metadata of the 58 variables listed in Supplementary Data 6, as described previously in ref. 13. The metadeconfoundR pipeline operates in the following manner: First naive associations are tested using Wilcoxon signed-rank tests (binary variables) or Spearman correlations (continuous variables). Resulting two-sided p-values are corrected with FDR. Variables influencing the same taxonomic feature were compared through nested likelihood ratio tests, and associations not reducible to any other variable were labeled as "deconfounded." Participant ID was included as a random variable to account for repeated measures. We considered metadata variables to be significantly associated with tested features when they were reported as deconfounded and if the absolute effect size was greater than 0.2 and the FDR was less than 0.05.

**Mediation analysis.** To assess whether changes in *Staphylococcus* abundance were mediated by sweat chloride levels in relation to ETI treatment duration, we performed a mediation analysis using the mediate function from the mediation R package[52] Linear models were fitted for the mediator and outcome, adjusting for sex and age. The mediation effect was estimated using the product-of-coefficients approach with nonparametric bootstrapping. Model assumptions were evaluated using the gvlma package[53].

**Procrustes analysis.** To assess cross-site compositional similarity, we performed Procrustes analyses on paired sputum-throat and sputum-stool samples using PCoA ordinations based on BC dissimilarity. Ordinations were generated using classical multidimensional scaling on the top 10 axes. The observed Procrustes sum of squares ($m^{12}$) for each site-pair was calculated using the protest() function from the vegan package[49] with 999 permutations.

To evaluate significance, we generated a null distribution by repeating the Procrustes analysis 1000 times on randomly selected, unpaired samples (excluding those used in the paired set), subsampled to match the size of the paired dataset. The observed $m^{12}$ statistic was compared to this null distribution to determine whether microbial compositions were more similar in paired samples than expected by chance.

**Detection of shared ASVs between habitats.** To assess microbial taxonomic overlap between habitats, ASV counts were transformed to presence/absence in the filtered dataset. This analysis was performed for paired and unpaired samples. For each sample-pair, ASVs present in both samples were identified, and the proportion of shared ASVs was calculated relative to the total ASV count in each sample. LMEs with ID included as a random effect were used to test whether paired samples shared significantly more ASVs than unpaired samples.

**IQR-Z-score transformation for CF-control comparison of taxa abundances.** To assess the deviation of microbial abundances in CF samples relative to our healthy control population, we employed an IQR-based Z-score transformation. This method standardizes the microbial abundances by expressing them in terms of their deviation from the median abundance of healthy controls, scaled by the interquartile range (IQR).

The IQR-based Z-score for each $taxon_i$ in a CF sample was calculated as follows:

$$Z_i = \frac{X_{CF,i} - \text{median}_i}{\text{IQR}_i}$$

where $X_{CF,i}$ is the abundance of taxon $i$ in the CF sample. A Z-score of 0 corresponds to no deviation from the median abundance observed in healthy controls for the $taxon_i$ in the specific CF sample. The calculated Z-scores were tested with LMEs as outlined above.

**Reporting summary**
Further information on research design is available in the Nature Portfolio Reporting Summary linked to this article.

## Data availability
All annotated sequencing data and clinical metadata are available at[54] (https://github.com/RebeccaLuise/IMMProveCF_public) with raw sequences accessible at (http://www.ncbi.nlm.nih.gov/bioproject/1080555) (PRJNA1080555).

## Code availability
All analysis scripts are available at[54] https://github.com/RebeccaLuise/IMMProveCF_public, https://doi.org/10.5281/zenodo.16878229.

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

## Acknowledgements

We express our gratitude to the individuals with CF and healthy controls, along with their families, for their participation in this study. We also appreciate the medical and nursing staff at the Children's Hospital Mainz, especially Verena Süß, for their support. Additionally, we thank

Elad Deiss-Yehiely for his thoughtful internal review of the manuscript. RLK was funded by the Deutsche Forschungsgemeinschaft (German Research Foundation - DFG), project number 551589343. SKFS and VHJD were funded by the DFG: Project ID 431232613 – SFB 1449 ("HYDROGEL"). VHJD was funded by the DFG grant "Integrative evolutionäre und ökologische Analyse von Antibiotikaresistenzen: Auftreten und Verbreitung vom bakteriellen Genom bis zur geographischen Landschaft" FO 1279/6-1. SKFS received funding from EU Horizon grant 101095540 ("IMMEDIATE"), Bundesministerium für Bildung und Forschung (BMBF) project 01EK2103A ("PROSPER") and BMBF project 01KI2404B ("JPIAMR-SEARCHER"). During this work's preparation, the authors used ChatGPT 4.0 to improve the text's legibility and accessibility. The authors reviewed and edited all content and take full responsibility for the final publication.

## Author contributions

RLK: Conceptualization, Methodology, Software, Validation, Formal analysis, Data Curation, Writing - Original Draft, Visualization, Project administration. MMB: Methodology, Software, Validation, Formal analysis, Data Curation, Visualization, Writing - Original Draft. ER: Investigation, Data Curation. LC: Investigation, Data Curation. LW: Investigation, Data Curation. KH: Methodology, Investigation, Data Curation, Project administration. NU: Methodology, Software, Validation, Formal analysis, Data Curation, Visualization, Writing - Original Draft. BH: Conceptualization, Validation, Writing - Review & Editing. TB: Software, Validation. TUPB: Software, Validation, Visualization. ON: Validation, Supervision. VHJD: Software, Validation, Supervision, Writing - Review & Editing. SL: Validation, Supervision, Writing - Review & Editing. SG: Validation, Resources, Supervision, Funding acquisition. LM: Conceptualization, Methodology, Validation, Resources, Supervision, Funding acquisition. KP: Conceptualization, Methodology, Validation, Resources, Supervision, Project administration, Funding acquisition. SKFS: Conceptualization, Methodology, Software, Validation, Writing - Review & Editing, Resources, Supervision, Funding acquisition.

## Funding

## Competing interests

Krystyna Poplawska served on the advisory board of Vertex Pharmaceuticals Inc. All other authors have no conflicts of interest to declare.

## Additional information

Rebecca Luise Knoll [1,2,3,4,5], Melanie Meihua Brauny [6,7,8], Evelyn Robert [1], Louisa Cloos [1], Lydia Waser [1], Katja Hilbert [1], Nina Ulmer [6,7,8], Barlo Hillen [9], Till Birkner [2,3,4], Theda Ulrike Patricia Bartolomaeus [2,3,4,10], Oliver Nitsche [1], Víctor Hugo Jarquín-Díaz [2,3,4], Susan Lynch [5], Stephan Gehring [1], Lisa Maier [6,7,8], Krystyna Poplawska [1] ✉ & Sofia Kirke Forslund-Startceva [2,3,4,10,11] ✉

[1]Children's Hospital, University Medical Center of the Johannes Gutenberg-University Mainz, Mainz, Germany. [2]Charité-Universitätsmedizin Berlin, Freie Universität Berlin and Humboldt-Universität zu Berlin, Berlin, Germany. [3]Max Delbrück Center for Molecular Medicine in the Helmholtz Association (MDC), Berlin, Germany. [4]Experimental and Clinical Research Center, Max Delbrück Center for Molecular Medicine and Charité-Universitätsmedizin Berlin, Berlin, Germany. [5]Division of Gastroenterology and Benioff Center for Microbiome Medicine, Department of Medicine, University of California San Francisco, San Francisco, CA, USA. [6]Interfaculty Institute of Microbiology and Infection Medicine, University of Tübingen, Tübingen, Germany. [7]Cluster of Excellence "Controlling Microbes to Fight Infections", University of Tübingen, Tübingen, Germany. [8]M3 Research Center, University Hospital Tübingen, Tübingen, Germany. [9]Department of Sports Medicine, Prevention, and Rehabilitation, Institute of Sports Science, Johannes Gutenberg-University Mainz, Mainz, Germany. [10]DZHK (German Centre for Cardiovascular Research), Berlin, Germany. [11]Structural and Computational Biology Unit, EMBL, Heidelberg, Germany. ✉e-mail: krpoplawska@icloud.com; sofia.forslund@mdc-berlin.de

