## [Transparent Peer Review file · Nature Communications]

CFTR modulator therapy drives microbiome restructuring through improved host physiology in cystic fibrosis: the IMMProveCF phase IV trial

Corresponding Author: Dr Rebecca Knoll

Version 0:

Reviewer comments:

Reviewer #1

(Remarks to the Author)

This substantial body of work describes characterization of the microbiome of people with CF (pwCF), following initiation of the revolutionary treatment CFTR modulator Elexacaftor-Tezacaftor-Ivacaftor (ETI). The study is strengthened by its use of multiple sample types (lower respiratory, oral swabs, and fecal samples) and longitudinal follow-up. This is the first study in this area to incorporate this breadth of sample type and analysis, and as such contributes a more holistic picture of the changes imparted due to ETI. The authors' transparent sharing of their bioinformatic code is also greatly appreciated and a major strength. I commend the authors for their study design and careful attention to adjusting for co-variables. I have a few specific comments to improve the clarity of the manuscript.

Major Comments:

- The age span of subjects in the study is broad (ages 6-55 years). Both the CF respiratory and intestinal microbiomes exhibit large compositional changes with age (e.g., *Pseudomonas* overtaking *S. aureus* as the primary respiratory pathogen as pwCF get older). I recognize that age is incorporated as a co-variate in analyses. I would recommend explicitly describing the impact of age on the microbiome throughout the manuscript. For example, I would consider moving lines 127-131 to earlier in that results section.
- From Lines 153-163 and Figure 2c, it is not clear if the increase *Pseudomonas* in sputum samples represents new detection of *Pseudomonas* in samples that were previously *Pseudomonas* negative. Figure 2c appears to indicate that *Pseudomonas* was not present or not dominant in baseline samples. I would recommend clarifying the description in the results section with a sentence such as “## Subjects provided baseline sputum samples, 11 of which provided at least one sputum sample after ETI”. (explicitly describe the intersection size Ext Fig 1 in the results section). And then a description of the *Pseudomonas* finding.
- What about *Pseudomonas* abundance in throat samples? This is particularly important since pwCF produce less sputum after ETI but appear to still harbor *Pseudomonas* in their sputum samples. Does *Pseudomonas* show up in the same throat samples from the same subject as sputum samples? How well do the culture results in Table 1 reflect the 16S rRNA sequencing? Can throat swabs be used as a proxy for sputum testing in clinical practice?
- The final concentrations of CFTR modulators used in bacterial culture experiments appears to be at least a hundred-fold more than is physiologically relevant (PMID 37491532, 32536510, 32044246). Furthermore, the testing of respiratory pathogens for susceptibility to ETI components or synergy of ETI with antibiotics is not particularly novel (PMID 27626100 & 36625583). The testing of non-pathogen respiratory microbiota members is more unique to this manuscript. Additional novelty could be added as suggested by the comment below.
- Similarly, from the pharmacokinetics of ETI, there is no known preferential distribution of this oral medication to the respiratory tract. We would expect that intestinal bacteria are also exposed to ETI. Was the in vitro effect of ETI on *E. coli* tested for example? This could be important since *Escherichia* is highlighted as changing in abundance after ETI initiation and particularly given the statement in Line 71.

- At times, the manuscript reads as a list of correlations noted rather than a coherent story. (e.g., Lines 267-270 Parabacteroides association with HgbA1c levels and corresponding citation from a non-CF population does not meaningfully contribute to the overall story. Another example (Lines 248-249), what is the proposed physiologic association between inhaled aztreonam or mannitol and changes to the gut microbiome? Systemic antibiotics (Line 252) seems much more plausible and aztreonam/mannitol are not depicted in Figure 3d). I appreciate the confounders tested and associations that were noted. But describing every clinical variable and medication's influence on the microbiome seems excessive and takes away from the main message of the manuscript.
- Other studies have determined changes in *S. aureus* abundance in response to ETI. Although mentioned in the intro, there is no discussion of how these new findings agree or disagree with previous studies.

Minor Comments:

- The wording in line 112 is confusing. I think this phrasing would be more appropriate "with a median reduction of up to 35 days compared to the participant's antibiotic exposures in the year prior to ETI initiation"
- Similarly, the wording in line 119 is confusing. Decreased by a median of 25 mg/g? or a range of decreases 25-223 mg/g?
- Line 119: calprotectin units should be microgram per gram, not milligram per gram
- Line 477: Nextera is misspelled.
- The letters (a-e) in the legend of Figure 3 are not consistent with the graphs. I believe panel A is missing its letter in the figure legend
- Discuss discrepancy with prior study showing sputum bacterial load decreases (PMID 36976651), contrary to what is shown in this manuscript
- The vertical axis in Figure 3d should not be italicized.

(Remarks on code availability)

We commend the authors for providing the code. This constitutes a usable resource for the community.

Reviewer #2

(Remarks to the Author)

Here the authors present a study of changes in respiratory and gut microbiomes on people with CF (pwCF) as they start and the progress on elexacaftor/tezacaftor/ivacaftor (ETI) therapy. The study follows 35 pwCF (including paediatric and adult patients, age range 6 to 55 years) from a single CF centre. They find that the CF respiratory and gut microbiota become more alike healthy control comparisons, but do not fully become 'healthy like'.

There are a growing number of single-centre ETI microbiome studies now published or being published, typically focusing on respiratory or gut separately. This manuscript examines effects of ETI therapy on both (respiratory – [spontaneously expectorated sputum and deep throat swabs] and gut [stool]).

A major criticism of this manuscript is that by combining respiratory and gut together that a lot of the finer detail presented in separate manuscripts on their own is lost here. Which is a shame. The advantage of combining would be if the authors were making distinct linkages between the respiratory and gut microbiomes (i.e. gut-lung axis). But that is not the case.

The manuscript would greatly benefit with the inclusion of a study limitations/caveats subsection in the discussion section. As presented the single-centre CF cohort is unusual as the 35 participants range from 6 to 55 years of age which would encompass a very large variation in living with the cumulative effects of CF (e.g. antibiotic exposure, lung damage, pancreatic insufficiency, etc). Being both a small paediatric and adult cohort the authors findings may well be specific to this cohort. Unusually (Line 417) all CF participants were recruited from a 'pediatric pulmonology outpatient clinic'.

Following which pwCF provided samples and when throughout the study is hard to follow. It would be beneficial to explain that more clearly to the reader within the manuscript.

Anecdotally, it is reported that the majority of pwCF who could produce sputum are no longer able to produce sputum once they commence ETI therapy. I say anecdotally as while this is widely accepted and noted, I have not seen that published as fact other than being a repeated line (as per here at Line 94). If the authors do have a reference, then please do add that. Those that can still do so are typically adults that have a mixture of severe lung disease and parenchymal damage. Therefore, the sputum analyses presented here (Line 92, '18 provided baseline sputum samples, with 11 contributing at least one follow-up sample....') would be of atypical / not representative of pwCF on ETI therapy. Therefore, this manuscript

would have to be rewritten/restructured acknowledge that. I suspect the sputum producers here are older patients (or with more severe disease) therefore it would be beneficial to describe who the sputum producers were.

Minor comments:

Line 89: 'up to 15 months post-treatment' this would these pwCF had finished treatment. Would on treatment be a better option.

Line 95 (and from above): Strongly suggest clarity on samples and patients is given. For example, how many longitudinal samples by samples type per participant.

Line 140-147: N = 11. SO was there enough sputum samples in longitudinal follow-up (at the 3month intervals) to do meaningful statistical analyses here?

Line 153: Suggest adding 'in this cohort' to read 'We further investigated taxonomic-level changes and found in this cohort a complete loss of Staphylococcus dominance'

Line 443: Please state what these kits were along with manufacturer details.

(Remarks on code availability)

Reviewer #3

(Remarks to the Author)

(Remarks on code availability)

Version 1:

Reviewer comments:

Reviewer #1

(Remarks to the Author)

The revised manuscript adequately addressed the reviewers' comments and the final manuscript is now suitable for publication.

One minor comment that the authors could consider is on the section describing the effects of the modulators on intestinal microbiota: The reduction in E. coli could be due to the reduction in inflammation which inadvertently changes the oxygen availability in the gut. This could be discussed briefly in the final version although it is not necessary for re-review.

(Remarks on code availability)

NA

Reviewer #2

(Remarks to the Author)

I appreciate that the authors have attempted to address my comments and that is appreciated.

However, the manuscript still has the fundamental flaws of being a small cohort from a single centre (with 35 patients ranging from 6 to 55 years) followed over a relatively short sampling period (considering ETI has been available for eligible people with CF of ≥ 12 years since ca. 2020 and ≥ 6 since ca. 2021). This study finds that both the respiratory and gut microbiota transition to being more healthy-like with time on ETI but do not achieve full healthy like status.

The same has been found in a number of similar recently published papers (e.g. typically single centre / small cohort / relatively short time frame) in both respiratory and gut microbiomes. For example: Respiratory microbiome: PMID: 37414422, PMID: 37841851, PMID: 36976651, PMID: 36154176, PMID: 38564694, PMID: 38916354, PMID: 34824018, PMID: 39406574, PMID: 37837613. Gut microbiome: PMID: 40483244, PMID: 38749891. A common conclusion from the majority of those papers is that future work needs to look at the long-term effects of ETI and on larger patient cohorts.

Additionally, there is still an over emphasis on the findings from unrepresentative sputum producing sub-population (n = 18, of which 11 provided follow-up samples [of which 9 provided more than 1 follow-up sample]).

(Remarks on code availability)

NA

Reviewer #3

(Remarks to the Author)

(Remarks on code availability)

Point-by-point response to editor and reviewer comments for:

Microbiome dynamics are mediated by changes in host physiology in people with cystic fibrosis undergoing CFTR-modulator therapy

Rebecca Luise Knoll^{1,2,3,4,5}, Melanie Meihua Brauny^{6,7,8}, Evelyn Robert¹, Louisa Cloos¹, Lydia Waser¹, Katja Hilbert¹, Nina Ulmer^{6,7,8}, Barlo Hillen⁹, Till Birkner^{2,3,4}, Theda Ulrike Patricia Bartolomaeus^{2,3,4,10}, Oliver Nitsche¹, Víctor Hugo Jarquín-Díaz^{2,3,4}, Susan Lynch⁵, Stephan Gehring¹, Lisa Maier^{6,7,8}, Krystyna Poplawska^{1**}, Sofia Kirke Forslund-Startceva^{2,3,4,10,11}

Reviewer #1:

Major Comments:

The age span of subjects in the study is broad (ages 6-55 years). Both the CF respiratory and intestinal microbiomes exhibit large compositional changes with age (e.g., Pseudomonas overtaking S. aureus as the primary respiratory pathogen as pwCF get older). I recognize that age is incorporated as a co-variate in analyses. I would recommend explicitly describing the impact of age on the microbiome throughout the manuscript. For example, I would consider moving lines 127-131 to earlier in that results section.

We thank the reviewer for this important point. Age is indeed a known contributor to microbiome variation in both healthy and CF populations. Our cohort's broad age range (6-55 years) introduces heterogeneity but also allows exploration of treatment responses across the CF lifespan.

To clarify, lines 127-131 of the original manuscript refer to associations between age and clinical variables, not microbiome composition.

Regarding microbiome analyses, we evaluated the influence of age and found it explained only a minor proportion of variance. Specifically, older individuals showed reduced sputum microbiome richness (observed ASVs, LME: Estimate = - 0.93, FDR < 0.05). However, in throat and stool samples, age was not significantly associated with any alpha or beta diversity measures (Supplementary Table 4).

We now clarify this in the main text (lines 106-111): "While age and sex significantly impacted several clinical parameters (as detailed below), their effect on microbiome composition was limited. Specifically, age was only associated with reduced microbial richness in sputum (N observed ASVs), with no significant impact observed on throat or stool alpha diversity but neither age nor sex affected throat or stool alpha diversity, nor overall microbial community structure (Bray-Curtis (BC) dissimilarity; PERMANOVA) across habitats. In contrast, ETI and other clinical factors drove significant improvements in both clinical parameters and microbiome composition compared to baseline (Fig. 1 and Extended Data Fig. 2)."

From Lines 153-163 and Figure 2c, it is not clear if the increase Pseudomonas in sputum samples represents new detection of Pseudomonas in samples that were previously Pseudomonas negative. Figure 2c appears to indicate that Pseudomonas was not present or not dominant in baseline samples. I would recommend clarifying the description in the results section with a sentence such as "## Subjects provided baseline sputum samples, 11 of which provided at least one sputum sample after ETI". (explicitly describe the intersection size Ext Fig 1 in the results section). And then a description of the Pseudomonas finding.

We thank the reviewer for this clarification request. We have updated the results section (lines 149-151): “Eighteen CF participants provided baseline sputum samples, 11 of whom provided at least one sputum sample after the start of ETI treatment (3 participants with 1 follow-up sample, 9 participants with more than 2 follow-up samples; Extended Data Fig. 1 a, b, 3f).”

To describe *Pseudomonas* findings more clearly we added from line 171-175: “Notably, 2 participants exhibited *Pseudomonas*-dominated sputum microbiota post-ETI initiation, neither of whom had *Staphylococcus* or *Pseudomonas* dominance at baseline. Additionally, 2 of the 6 participants who were initially *Staphylococcus*-dominated showed an increased relative abundance of *Pseudomonas* post-treatment, although without reaching dominance (Extended Data Fig. 3f).”

What about Pseudomonas abundance in throat samples? This is particularly important since pwCF produce less sputum after ETI but appear to still harbor Pseudomonas in their sputum samples. Does Pseudomonas show up in the same throat samples from the same subject as sputum samples? How well do the culture results in Table 1 reflect the 16S rRNA sequencing? Can throat swabs be used as a proxy for sputum testing in clinical practice?

We thank the reviewer to mention this important point, especially given the reduced sputum production following ETI, which limits lower airway sampling. We assessed whether throat swabs could serve as a practical proxy for sputum in detecting *Pseudomonas*, using both culture and 16S rRNA data from paired samples.

In 45 paired samples analyzed by 16S, *Pseudomonas* detection in throat swabs showed 80% accuracy, 87% sensitivity, and 77% specificity relative to sputum. Culture-based comparisons were similar (accuracy 82%, sensitivity 85%, specificity 81%), with only 2 of 13 *Pseudomonas*-positive sputum samples missed by throat swabs.

These results support the potential utility of throat swabs for *Pseudomonas* surveillance. We have now highlighted this result more clearly in the manuscript (lines 321-324, Extended Data Fig. 7i-j), and summarized these comparisons in Supplementary Table 15. Supplementary Table 15 extends the culture results shown in Table 1 by providing detailed performance metrics of 16S rRNA-based detection relative to culture-based detection across both sample types.

The final concentrations of CFTR modulators used in bacterial culture experiments appears to be at least a hundred-fold more than is physiologically relevant (PMID 37491532, 32536510, 32044246) . Furthermore, the testing of respiratory pathogens for susceptibility to ETI components or synergy of ETI with antibiotics is not particularly novel (PMID 27626100 & 36625583). The testing of non-pathogen respiratory microbiota members is more unique to this manuscript. Additional novelty could be added as suggested by the comment below.

We thank the reviewer for noting this discrepancy. We apologize for the confusion in our original methods section. The concentrations listed (16 mM–1.25 mM) referred to stock solutions, not final tested concentrations.

The actual concentrations used in culture assays were:

- CFTR modulators: 160 μ M to 1.25 μ M
- Antibiotics: 20 μ M to 2.5 μ M

Accordingly, we have clarified this in the methods section (lines 591-592). IC25 values are reported at the concentration where inhibition was observed, as depicted in Figure 2 and 3g, which we revised for better clarity as well.

Published data show serum concentrations of ivacaftor (IVA~3.5 μM), elexacaftor (ELX~20.5 μM), and tezacaftor (TEZ~10.4 μM) in treated individuals (PMID: 34668357), with IVA sputum levels up to ~0.5 μM (2.5 h post-dose in a single individual, PMID: 27792891). Our IC25 values for *S. aureus* (~10 μM) and lower values for some commensals fall within or near this range. Although we did not measure drug levels directly, local accumulation in tissues has been reported (PMID 36625583), supporting biological plausibility.

As the reviewer noted, the inclusion of non-pathogenic respiratory microbiota members in susceptibility testing represents a novel aspect of this study and, to our knowledge, has not been systematically addressed in the field before.

Similarly, from the pharmacokinetics of ETI, there is no known preferential distribution of this oral medication to the respiratory tract. We would expect that intestinal bacteria are also exposed to ETI. Was the in vitro effect of ETI on E. coli tested for example? This could be important since Escherichia is highlighted as changing in abundance after ETI initiation and particularly given the statement in Line 71.

We appreciate this insightful suggestion. To test whether the observed reduction in *Escherichia-Shigella* could result from direct drug effects, we performed susceptibility testing of *E. coli* and other gut microbiota members to CFTR modulators. *E. coli* showed the lowest susceptibility among tested taxa, suggesting its decline is more likely due to indirect effects, such as reduced inflammation, decreased antibiotic exposure, or broader ecological shifts. We now include these results in Figure 3g and describe them in the revised results section (lines 279-282): “Following up on this, we investigated whether the reduction of *Escherichia-Shigella* might be attributed to a direct antibacterial activity of any of the CFTR-modulators and tested drug susceptibility in our *in vitro* direct drug-bacteria interaction assay. *E. coli* was not directly affected by the modulators, although several commensal gut bacteria were (Fig. 3g).”

As the reviewer suggested, extending this approach to gut-associated taxa such as *E. coli* further underscores the novelty of investigating potential direct effects of CFTR modulators on non-pathogenic members of the microbiota.

At times, the manuscript reads as a list of correlations noted rather than a coherent story. (e.g., Lines 267-270 Parabacteroides association with HgbA1c levels and corresponding citation from a non-CF population does not meaningfully contribute to the overall story. Another example (Lines 248-249), what is the proposed physiologic association between inhaled aztreonam or mannitol and changes to the gut microbiome? Systemic antibiotics (Line 252) seems much more plausible and aztreonam/mannitol are not depicted in Figure 3d). I appreciate the confounders tested and associations that were noted. But describing every clinical variable and medication’s influence on the microbiome seems excessive and takes away from the main message of the manuscript.

We thank the reviewer for this thoughtful critique. We agree that listing every observed association may dilute the central narrative. Accordingly, we have removed the sentence regarding *Parabacteroides* and HgbA1c.

Regarding inhaled aztreonam and mannitol: although not systemically administered, both were included in our analysis due to plausible, albeit indirect, effects on the gut microbiome. Aztreonam, although inhaled, is an antibiotic, and may exert off-target effects through mucosal absorption or swallowing of residual drug. Mannitol, a polyol with osmotic properties, is known from nutritional studies to reach the gastrointestinal tract where it can influence the microbiota, such as by promoting *Bifidobacterium* growth or causing laxative effects in healthy individuals (PMID: 30721958). While speculative, these mechanisms provide a possible biological rationale for including them in exploratory analysis. Still, we agree this level of detail detracted from the main narrative, and have streamlined the results to emphasize only the most relevant findings. We have therefore restructured the results and discussion to emphasize the variables with the strongest associations, namely *months from treatment start, mannitol inhalation, number of systemic antibiotic treatment days and oral corticosteroids* (Lines 265-268). For readers interested in these exploratory associations, they remain accessible in Extended Data Figure 5c.

Other studies have determined changes in S. aureus abundance in response to ETI. Although mentioned in the intro, there is no discussion of how these new findings agree or disagree with previous studies.

We thank the reviewer for this important point and have now added a dedicated section to the Discussion to address it directly (lines 379-384): “Multiple cohort studies have reported a general decline in *Staphylococcus* and *Pseudomonas* burden following ETI initiation (PMID: 36976651, PMID: 38036261, PMID: 37414422). However, many patients remain colonized by the same pathogens post-treatment. Interestingly, one study observed no significant change in either organism and even noted an increase in *Pseudomonas* detection by culture (PMID: 34824018). These findings point to host- and pathogen (strain)-specific factors as likely contributors to the variable response to ETI treatment.”

This comparison also prompted us to reflect more critically on why we did not observe a decrease in *Pseudomonas* in our cohort. We have now added to the discussion (lines 409-412): “This persistence may, in part, reflect cohort-specific differences; for instance, our sputum follow-up cohort included a greater proportion of older patients with more advanced lung disease, which are factors previously associated with reduced pathogen clearance (PMID: 38036261).”

This addition also addresses Reviewer #2’s request to include a clear statement regarding the limited generalizability of our cohort, which we have now included in lines 447-451: “While our cohort size enables robust analyses, generalizability remains limited due to heterogeneity in age, disease severity, and treatment regimens, resulting in smaller subgroups for certain therapies. Future studies should validate these findings in larger, more stratified CF populations to enhance robustness and applicability.”

Minor Comments:

We thank the reviewer for these helpful suggestions and have addressed each point as follows:

- We have revised the phrasing in lines 114 and 119 to improve clarity, as suggested.
- Calprotectin units have been corrected to $\mu\text{g/g}$.
- The spelling of “Nextera” in line 522 has been corrected.
- The panel lettering in Figure 3 has been updated to ensure consistency between the figure and legend.
- The vertical axis label in Figure 3d has been reformatted to remove italicization.
- Regarding the study reporting decreased sputum bacterial load (PMID: 36976651), we appreciate the reviewer bringing this to our attention. Upon closer examination,

we note that our findings are in fact consistent with this work: while certain pathogens decreased, the total bacterial load as measured by 16S rRNA gene ddPCR remained stable (see Supplemental Figure 12E of that study).

Reviewer #2 :

A major criticism of this manuscript is that by combining respiratory and gut together that a lot of the finer detail presented in separate manuscripts on their own is lost here. Which is a shame. The advantage of combining would be if the authors were making distinct linkages between the respiratory and gut microbiomes (i.e. gut-lung axis). But that is not the case.

We thank the reviewer for this thoughtful and constructive comment. We agree that both the respiratory and gut datasets are rich enough to support independent publications. However, by aligning analyses of these two systems within a unified statistical framework, we cannot only compare their treatment responses directly but also distinguish how the mechanisms shaping each habitat may differ. This integrative approach allows us to disentangle compartment-specific from shared effects of ETI treatment with greater precision and statistical stringency than would be possible in two separate, unlinked analyses. Their juxtaposition provides the necessary context to interpret shifts in each environment relative to the other—insights that would likely be lost or obscured if the systems were studied in isolation.

In the original manuscript, we described distinct temporal patterns: while respiratory microbiomes (throat and sputum) shift early and stabilize, the stool microbiome responds more gradually and remains more variable. The taxa driving these shifts also differ by habitat: *Staphylococcus* reductions in sputum associate with improved CFTR function (sweat chloride), while *Escherichia* declines in stool link to reduced intestinal inflammation (fecal calprotectin).

We acknowledge that we initially underemphasized cross-habitat analyses. In response, we have now expanded these analyses to directly explore the gut-lung axis in comparison to the respiratory axis (lines 307-329, Extended Data Fig. 7), which also helps to contextualize and scale the similarity of the different respiratory habitats. Our expanded analyses include:

- **Bray-Curtis PCoA and Procrustes analyses**, showing strong habitat-specific clustering, with particularly high intra-individual similarity between throat and sputum samples—suggesting coordinated respiratory responses.
- **Marked dissimilarity between stool and respiratory samples**, with no evidence of synchronized treatment effects across the gut-lung axis.
- **Higher ASV sharing and correlated alpha diversity** between throat and sputum samples, underscoring respiratory connectivity.
- **Clinical relevance of respiratory linkage**, as throat swabs predict sputum presence of pathogens such as *Pseudomonas* with high sensitivity and specificity.

Additionally, we refined the interpretation of Figure 4c, which already demonstrated synchronized microbial shifts in the respiratory tract not observed in stool-respiratory comparisons. This section has been reworded for clarity (lines 299-303).

Finally, while 16S rRNA gene sequencing limits taxonomic and functional resolution, especially in low-biomass respiratory samples, we performed all feasible analyses within these constraints. The expanded integrative analyses now better demonstrate the value of our systems-level approach.

The manuscript would greatly benefit with the inclusion of a study limitations/caveats subsection in the discussion section. As presented the single-centre CF cohort is unusual as the 35 participants range from 6 to 55 years of age which would encompass a very large variation in living with the cumulative effects of CF (e.g. antibiotic exposure, lung damage, pancreatic insufficiency, etc). Being both a small paediatric and adult cohort the authors findings may well be specific to this cohort. Unusually (Line 417) all CF participants were recruited from a 'pediatric pulmonology outpatient clinic'.

We appreciate the reviewer's thoughtful suggestion to more clearly articulate the study's limitations. While our single-center cohort spans a wide age range, we view this both as a limitation and a strength. It introduces heterogeneity in disease severity and treatment exposure, factors we accounted for using confounder-aware univariate and multivariate statistical approaches, ensuring that only robust, non-confounded associations are reported. In this, age was explicitly included as a covariate and found to have limited impact on key outcomes.

In our view, recruiting all participants from a single center is a strength of the study design from a perspective of testing for an intervention effect. It allows consistent phenotyping and biosampling which can be challenging in multicenter studies, where otherwise site effects combined with imbalanced sampling of baseline severity between sites can result in spurious findings. This consistency is particularly valuable for low-biomass respiratory samples such as these, where additionally site-specific microbial contamination substantially can confound results and be difficult to disentangle from genuine patient population differences. Similarly, the relatively homogeneous disease severity among the participants (all exhibiting some degree of pancreatic insufficiency and at least one F508del risk mutation) serves to increase statistical power relative to the size of the cohort. However, future validation studies on different representative cohorts are called for to further increase confidence in generalizability, and we now highlight this more clearly in the discussion.

In response to the reviewer's comment, we have expanded the limitations section (lines 447-451): "While our cohort size enables robust analyses, generalizability remains limited due to heterogeneity in age, disease severity, and treatment regimens, resulting in smaller subgroups for certain therapies. Future studies should validate these findings in larger, more stratified CF populations to enhance robustness and applicability."

Regarding the clinical setting, while it may seem unusual that adults were followed in a pediatric clinic, this reflects standard CF care in Germany, where pediatric CF specialists often continue care into adulthood within university medical centers. We have clarified this in the manuscript (lines 459-461).

Following which pwCF provided samples and when throughout the study is hard to follow. It would be beneficial to explain that more clearly to the reader within the manuscript.

Thank you for this suggestion. We have now added Extended Data Fig. 1a to clarify which participants contributed which sample types at which timepoints. This figure distinguishes three groups: (i) sputum at baseline and follow-up, (ii) sputum at baseline only, and (iii) no sputum. Extended Data Figs. 1b-d provide intersection sizes over time. We also now direct readers to these new figures and clarify sputum subgroup structure in the main text (lines 93-97).

Anecdotally, it is reported that the majority of pwCF who could produce sputum are no longer able to produce sputum once they commence ETI treatment. I say

anecdotally as while this is widely accepted and noted, I have not seen that published as fact other than being a repeated line (as per here at Line 94). If the authors do have a reference, then please do add that.

We agree that this phenomenon, while widely reported, is rarely formally quantified. We therefore consider here reporting on it quantitatively to be a minor additional advance. In addition we now cite published data supporting this observation:

- In a recent study (PMID: 36976651), 89% of participants could produce sputum at baseline, dropping to 53.5% after 6 months of ETI. Sputum induction failed in 10% of patients.
- Registry data from 1,092 pwCF in Germany show sputum-based cultures fell from 70.6% pre-ETI to 42.8% post-ETI (PMID: 38036261).

We have added both references to support the statement made at line 94.

Those that can still do so are typically adults that have a mixture of severe lung disease and parenchymal damage. Therefore, the sputum analyses presented here (Line 92, '18 provided baseline sputum samples, with 11 contributing at least one follow-up sample....') would be of atypical / not representative of pwCF on ETI treatment. Therefore, this manuscript would have to be rewritten/restructured acknowledge that. I suspect the sputum producers here are older patients (or with more severe disease) therefore it would be beneficial to describe who the sputum producers were.

We appreciate the reviewer's suggestion to better characterize the subgroup of patients providing sputum samples at baseline and follow-up. As suspected, sputum producers were generally older with lower lung function. We have now included a detailed comparison of cohort characteristics for the three subgroups (no sputum, baseline only, baseline plus follow-up) via Table 2 (added for reference below) and discuss this heterogeneity more explicitly in the manuscript (lines 95-97, 106-111, 409-412, 447-451).

While this subgroup is not representative of all pwCF on ETI, it represents a clinically meaningful population: individuals with persistent, severe lung disease despite therapy. We see this not only as a limitation but rather as an opportunity, capturing the transition of pwCF from pre-ETI disease states toward post-ETI phenotypes. This continuum of response is central to our longitudinal design and offers rare, timely insight as untreated pwCF populations become increasingly harder to sample as ETI becomes standard of care. Moreover, all features differing between sputum producers and non-producers (e.g., age, lung function) were explicitly tested and incorporated as covariates in relevant analyses, ensuring these factors were accounted for when interpreting microbial and clinical outcomes.

Thus, while not representative of all ETI-treated pwCF, this cohort is representative of pwCF initiating ETI from varied, clinically relevant baselines, allowing investigations into causes of heterogeneity of treatment response.

Table 2: Cohort Characteristics at baseline between subgroups: stratified by sputum availability and follow-up status

FU = follow-up; **Spu FU** = sputum sample with follow-up; **Spu NFU** = sputum sample without follow-up; **NoSpu** = no sputum sample available; **N** = number of participants; **n (%)** = number and percentage of participants; **IQR** = interquartile range; **ppFEV1** = percent predicted forced expiratory volume in 1 second; **ppFVC** = percent predicted forced vital capacity. Statistical tests used: **FT** = Fisher's exact test, **KW** = Kruskal-Wallis test, **W** =

Wilcoxon rank-sum test for post-hoc pairwise comparisons where applicable. NA = not applicable (no sputum sample available for microbiological assessment).

Characteristics at Baseline	Sputum with FU N = 11 ¹	Sputum no FU N = 7 ¹	No Sputum N = 16	Statistics
Sex				
Female n (%)	5 (45%)	5 (71%)	9 (56%)	FT: $p = 0.38$
Age in years				
Median (IQR)	26(23,32)	28 (21, 30)	16 (9, 28)	KW: $\chi^2=4.5$, $p=0.11$
Age groups				
≥ 20 n (%)	10 (91%)	5 (71%)	8 (50%)	FT: $p = 0.03$; Spu NFU - NoSpu: $p = 0.17$; Spu FU - NoSpu: $p = 0.03$; Spu FU - Spu NFU: $p = 0.35$
12-19 n (%)	1 (9.1%)	1 (14%)	1 (6.3%)	
≤ 11 n (%)	0	1 (14%)	7 (44%)	
Mutation				
F508del homozygous n (%)	9 (82%)	5 (71%)	12 (75%)	FT: $p = 0.7$
Lung function				
ppFEV1 Median (IQR)	69 (52, 76)	79 (69, 83)	94 (82, 99)	KW: $\chi^2=6.5$, $p = 0.04$; Spu NFU - NoSpu W: $p = 0.18$; Spu FU - NoSpu W: $p = 0.02$; Spu FU - Spu NFU W: $p = 0.23$
ppFVC Median (IQR)	84 (80, 96)	94 (91, 100)	100 (93, 115)	KW: $\chi^2=4.2$, $p=0.12$
Sweat chloride				
[mmol/l] Median (IQR)	85 (77, 95)	97.2 (88.5, 99.4)	82 (78, 93)	KW: $\chi^2=4.8$, $p=0.09$
Bacterial culture results in sputum				
Staphylococcus aureus pos. n (%)	7 (64%)	4 (57%)	NA	FT: $p = 1$
Pseudomonas aeruginosa pos. n (%)	7 (64%)	3 (43%)	NA	FT: $p = 0.6$

* 1 patient (IMP8) provided a single sputum sample at Visit 9, but no baseline sample, and is not accounted for in this subgroup analysis.

Minor comments:

Line 89: Changed to: "...contributed samples limited to the first 15 months of treatment."

Line 95: Addressed through improved sample overview (see Major Comment 3).

Line 140-147: We acknowledge the small sample size (N=11), but linear mixed-effects models are appropriate for repeated, unbalanced measures and still yielded statistically significant results, supporting their validity.

Line 153: Modified as suggested to read: "...found in this cohort a complete loss of *Staphylococcus* dominance..." (now line 165)

Line 443: We added a reference to a previous publication where we detailed this information (now line 487).

REVIEWERS' COMMENTS

Reviewer #1 (Remarks to the Author):

The revised manuscript adequately addressed the reviewers' comments and the final manuscript is now suitable for publication.

One minor comment that the authors could consider is on the section describing the effects of the modulators on intestinal microbiota: The reduction in *E. coli* could be due to the reduction in inflammation which inadvertently changes the oxygen availability in the gut. This could be discussed briefly in the final version although it is not necessary for re-review.

We thank the reviewer for this positive feedback. We highly appreciate the reviewer's idea to shortly comment on the mechanisms by which facultative anaerobes are adapted to the inflammatory environment.

ACTION TAKEN: We added in line 438-442:

Escherichia, as a facultative anaerobe, thrives in inflammatory environments by tolerating higher oxygen, using respiratory electron donors and epithelial-derived nutrients like ethanolamine (PMID 26185088). Further, *Escherichia* has a high capacity to resist host-derived antimicrobials and antibiotics. Therefore, reduced inflammation and antibiotic use during ETI likely disadvantaged *Escherichia*, facilitating microbiome recovery.

Reviewer #2 (Remarks to the Author):

I appreciate that the authors have attempted to address my comments and that is appreciated.

However, the manuscript still has the fundamental flaws of being a small cohort from a single centre (with 35 patients ranging from 6 to 55 years) followed over a relatively short sampling period (considering ETI has been available for eligible people with CF of ≥ 12 years since ca. 2020 and ≥ 6 since ca. 2021). This study finds that both the respiratory and gut microbiota transition to being more healthy-like with time on ETI but do not achieve full healthy like status.

We thank the reviewer for highlighting the limitations of cohort size, single-center design, and follow-up duration.

ACTION TAKEN: We now underscore these constraints more explicitly in the Discussion and Limitations sections. In particular, we revised lines 465–469 to state:

“Although our sample size is modest, the study design allows robust statistical inference within the studied population, including FDR control (<0.05), which minimizes the risk of spurious findings. Nonetheless, the single-center design and smaller subgroup sizes limit generalizability to populations with different clinical characteristics.”

We also emphasize that while long-term follow-up will be essential, existing evidence and our own results indicate that the majority of microbiome changes occur within the first 6 months of therapy, making our timeframe sufficient to capture the distinct kinetics of gut and airway responses. We now explicitly highlight the exciting possibility that further changes may occur on longer timescales, and point to this as an important direction for future work in our conclusion (lines 470-484):

“Larger, multi-center cohorts with extended follow-up and integrated metagenomic, multi-omic, and longitudinal designs will be critical to validate and expand upon these findings. Given the heterogeneity of CF and especially the CF lung microbiome, such efforts will be particularly valuable for capturing the full spectrum of sub-presentations and populations (e.g., *Pseudomonas*- or *Staphylococcus*-colonized vs. non-colonized, sputum producers vs. non-producers), and for disentangling the diverse factors shaping host–microbiome interactions in this disease. Moreover, while our study includes up to 24 months of follow-up, it remains possible that later changes or further normalization toward healthy states may emerge only with longer-term observation, providing an exciting opportunity for future investigation. Despite these limitations, our study, through multi-site (respiratory and gut) sampling, a comparatively large matched healthy control group, and a statistical framework accounting for confounding and mediation, provides essential groundwork for future large-scale investigations. Overall, this work represents a critical step in understanding how ETI therapy modulates microbiome–host interactions in CF across different habitats and highlights the potential for reversing dysbiosis through improved host physiology.”

The same has been found in a number of similar recently published papers (e.g. typically single centre / small cohort / relatively short time frame) in both respiratory and gut microbiomes. For example: Respiratory microbiome: PMID: 37414422, PMID: 37841851, PMID: 36976651, PMID: 36154176, PMID: 38564694, PMID: 38916354, PMID: 34824018, PMID: 39406574, PMID: 37837613. Gut microbiome: PMID: 40483244, PMID: 38749891. A common conclusion from the majority of those papers is that future work needs to look at the long-term effects of ETI and on larger patient cohorts.

We thank the reviewer for this feedback. To address the reviewer’s comment regarding similarities with previously published work, we created Supplementary Table S1 comparing our study with similar studies published to date (including those the reviewer kindly mentioned). Most are single-center with limited follow-up and focus exclusively on either the respiratory or gut microbiome. Notably, our study is the first to combine multi-site sampling with substantial longitudinal follow-up (up to 24 months), which is what lets us contrast gut and airway microbiome dynamics without introducing confounding from combining multiple datasets each with their own biases. We also applied a robust statistical framework controlling for confounding and mediation effects within the study itself, and included a comparatively large, age- and sex-matched healthy control cohort, exceeding most previous studies, which often included few or no controls.

ACTION TAKEN: To ensure this comparison and context is not lost, we now reference Supplementary Table S1 and the literature it spans explicitly in the Introduction (line 60-63):

“Several recent studies have reported similar microbiome transitions following ETI therapy (Supplementary Table S1), although most were single-center with short follow-up and focused on either the respiratory or gut microbiome in isolation.”

Additionally, there is still an over emphasis on the findings from unrepresentative sputum producing sub-population (n = 18, of which 11 provided follow-up samples [of which 9 provided more than 1 follow-up sample]).

We appreciate the reviewer’s concern regarding the representativeness of the sputum subpopulation.

ACTION TAKEN: In our revision, we clarified this limitation by revising lines 395–399 from:

“Within the sputum microbiome, our findings revealed a reduction in *Staphylococcus* following ETI therapy.”
to:

“Despite the modest follow-up sample size, sputum samples from the studied population clearly showed a statistically significant (FDR < 0.05) reduction in *Staphylococcus* following ETI therapy. We note that sputum producers represent a subset of people with CF who may differ clinically from non-producers (Table 2), and therefore this subgroup may not fully capture the spectrum of lung microbiome treatment responses.”

We further emphasized this limitation in our conclusion, noting that larger cohorts of sputum producers and broader colonization profiles will be needed to draw more generalizable conclusions (as highlighted above under Comment 1).

We thank the reviewer for these constructive suggestions, which have strengthened the manuscript by clarifying generalizability, highlighting study advances, and emphasizing the novelty of our multi-site, longitudinal approach.

Supplementary table S1: CF microbiome and ETI studies

PMID	First author (year)	Study design & follow-up	Sample type(s)	Has healthy controls	Notes / comparison to our study	Cited
3741 4422	Schaupp (2023)	Single center, larger initial cohort but high dropout (n=65 → n=28 at 12 months), 12-month follow-up	Sputum	Yes, (n=10)	P. aeruginosa reduction at 3, but not at 12 months, healthy levels not reached, shorter follow-up; larger cohort	Yes
3784 1851	Martin (2023)	Single center; 7 pwCF on ETI, 9 pwCF no ETI; only 2 individuals followed >1 year	Sputum	No	Very small sample; minimal long-term data	No
3697 6651	Nichols (2023)	Multi-center PROMISE study, 236 pwCF, 6-month follow-up	Sputum	No	PCR and culture methods, no microbiome profiling; most participants remain infected with pathogens	Yes
3856 4694	Armbruster (2024)	Single center, 16 pwCF, only one post-ETI sample at max 12-month follow-up	Sinus, throat	No	Only one follow-up sample per person; Persistence and evolution of P. aeruginosa	Yes
3891 6354	Hilliam (2024)	Single center, 38 adults but only 21 post-ETI participants, 12-months max follow-up	Sinus, Sputum	No	Interesting sinus dynamics; P. aeruginosa and S.aureus persist in sinuses; limited long-term data	No
3482 4018	Sosinski (2022)	24 adults, one single sample after early ETI initiation	Sputum	No	Small data set, first study in the field	Yes
3783 7613	Zemke (2024)	23 adults; one follow-up at median 9 months post-ETI	Sinus	No	No change in Staphylococcus ; limited longitudinal data	No
3940 6574	Steinberg (2025)	20 adolescents, 3-month follow-up	Throat	No	Very short follow-up	No
3615 4176	Pallenberg (2022)	Single center, 31 pwCF, 11-month follow-up	Throat, (Sputum at baseline)	No	Reduction of S. aureus and P. aeruginosa	Yes
4048 3244	Duong (2025)	Multi-center PROMISE study;	Stool	No	Large cohort but limited follow-up;	No* *very

		124 participants; 1 and 6-month post-ETI stool samples			supports our E. coli and calprotectin data	recently published
3874 9891	Marsh (2024)	20 pwCF; samples at 3, 6, and >17 months of ETI treatment	Stool	Yes, (n=10)	Moderate follow-up; smaller sample size	Yes